# Dorsoventral-mediated *Shh* induction is required for axolotl limb regeneration

**Sakiya Yamamoto[1], Saya Furukawa[1], Ayaka Ohashi[1], Mayuko Hamada[1,2], Akira Satoh[1]***

[1]Okayama University, Graduate School of Environmental, Life, Natural Science and Technology, Okayama, Japan; [2]Ushimado Marine Institute (UMI), Okayama University, Okayama, Japan

## eLife Assessment

This **fundamental** work by Yamamoto and colleagues advances our understanding of how positional information is coordinated between axes during limb outgrowth and patterning. They provide **convincing** evidence that the dorsal-ventral axis feeds into anterior-posterior signaling, and identify the responsible molecules by combining transplantations with molecular manipulations. This work will be of broad interest to regeneration, tissue engineering, and evolutionary biologists.

***For correspondence:**
satoha@cc.okayama-u.ac.jp

**Competing interest:** The authors declare that no competing interests exist.

**Abstract** Axolotls (*Ambystoma mexicanum*) exhibit a remarkable ability to regenerate limbs. Classical experiments have suggested that contact between cells derived from distinct orientations—dorsal, ventral, anterior, and posterior—within the regenerating blastema is necessary for accurate limb pattern formation. However, the molecular basis for this requirement has remained largely unknown. Here, we demonstrate that both dorsal and ventral tissues are required for limb formation via induction of *Shh* expression, which plays a crucial role in limb patterning. Using the accessory limb model, we induced position-specific blastemas lacking cells derived from a single orientation (anterior, posterior, dorsal, or ventral). Limb patterning occurred only in blastemas containing both dorsal- and ventral-derived cells. We further observed that *Shh* expression requires dorsoventral contact within a blastema, highlighting the necessity of dorsoventral contact for inducing *Shh* expression. Additionally, we identified WNT10B and FGF2 as dorsal- and ventral-mediated signals, respectively, that create the inductive environment for *Shh* expression. Our findings clarify the role of dorsal and ventral cells in inducing *Shh*, a mechanism that has rarely been studied in the context of limb regeneration and pattern formation. This model provides new insights into how cells with different positional identities drive the regeneration process.

## Introduction

Axolotls (*Ambystoma mexicanum*) possess remarkable regenerative abilities, enabling the regeneration of entire limbs after amputation. Among tetrapods (four-limbed vertebrates), only urodele amphibians retain the lifelong ability to regenerate fully developed limbs. Investigating the molecular mechanisms underlying axolotl limb regeneration could provide valuable insights for advancing regenerative medicine in humans, potentially leading to new therapies for tissue repair and organ regeneration.

The limb regeneration process can be divided into two stages: the blastema (regenerating limb primordium) induction process and the limb patterning process. The induction of a blastema, which contains highly proliferative multipotent and unipotent cells, depends on the presence of nerves at the injured region; when a denervated limb is amputated, a blastema is not induced (***Todd, 1823***). This

blastema induction process is followed by the limb patterning process. The limb patterning process has been considered to be mainly a recapitulation of limb development. Actually, activation of many developmental genes can be found during this phase. Such sequential events accomplish axolotl limb regeneration.

To accomplish limb patterning, cells need to know their address within a limb and exert their position-specific role. Such addresses of cells have been referred to as positional identities. In limb development/regeneration, positional identities are thought to be established along three-dimensional axes—anteroposterior, dorsoventral, and proximodistal. In amniote limb development, WNT7A, secreted from dorsal ectoderm, induces *Lmx1b* expression in the underlying mesenchyme, thereby specifying dorsal identity (*Riddle et al., 1995*; *Vogel et al., 1995*; *Cygan et al., 1997*; *Chen et al., 1998*; *Chen and Johnson, 2002*). On the ventral side, the ventral identity is specified indirectly by *En1*, which is expressed in the ventral ectoderm, restricts *Wnt7a* expression to the dorsal ectoderm and thereby prevents induction of *Lmx1b* in the ventral mesenchyme (*Loomis et al., 1996*; *Logan et al., 1997*; *Chen and Johnson, 2002*). Similarly, SHH is known to regulate anteroposterior patterning (*Riddle et al., 1993*). In axolotl limb regeneration, the establishment of the dorsoventral positional identities is still largely unknown, except for dorsal-specific *Lmx1b* expression (*Shimokawa et al., 2013*). Regarding the anteroposterior axis, *Shh* expression is highly conserved and restricted to the posterior margin. Though the process of the establishment of the positional identities is still veiled in axolotl limb regeneration, it is apparent that the regeneration blastema possesses regional specificity.

For successful limb patterning in limb regeneration, interactions between anterior, posterior, dorsal, and ventral cells within a blastema have been considered essential. The importance of these interpositional interactions was investigated in creating double-half limbs. Amputating the double-half limbs can provide a blastema in a 'one-positional-identity-missing' state. For example, when a double-dorsal limb—a chimeric limb surgically generated by excising the ventral half and replacing it with a dorsal half from the contralateral limb while preserving anteroposterior orientation—is amputated, the resulting blastema lacks ventral positional identity. Such blastemas lacking one positional identity can form hypomorphic, spike-like structures or fail to regenerate; in other cases, they regenerate limbs with complete anteroposterior axes accompanied by symmetric dorsoventral duplications (*Bryant, 1976*; *Bryant and Baca, 1978*; *Burton et al., 1986*). It is noteworthy that, in the non-regenerating cases, structural patterns along the anteroposterior axis appear to be lost even though both anterior and posterior cells should, in principle, be present in a blastema induced from a double-half limb. These observations are consistent with the idea that, during axolotl limb regeneration, signals mediated by dorsal cells must act on ventral cells, and signals mediated by ventral cells must act on dorsal cells, during the limb patterning process. In contrast, in amniote limb development, *Wnt7a/Lmx1b* or *En1* mutants show that limbs can exhibit anteroposterior patterning even when tissues are dorsalized or ventralized—that is, in the relative absence of ventral or dorsal cells, respectively (*Riddle et al., 1995*; *Chen et al., 1998*; *Loomis et al., 1996*). Taken together, these findings indicate that the mechanism underlying limb patterning during axolotl regeneration differs from that operating during amniote limb development. Additional experiments involving ectopic contact between opposite regions (e.g., anterior–posterior or dorsal–ventral) of a blastema further support the critical role of interpositional interactions in the limb patterning process, as such interactions can lead to ectopic limb formation (*Iten and Bryant, 1975*; *Bryant and Iten, 1976*; *Maden, 1980*; *Stocum, 1982*). These findings suggest that signals mediated by each positional identity are required for successful limb patterning, such that the absence of any one of them leads to failure. To further understand limb regeneration, the molecular basis of these positional identity-mediated signals in limb patterning process should be elucidated.

The accessory limb model (ALM), a non-amputation experimental system, provides a valuable approach for studying axolotl limb regeneration, particularly the role of interpositional interactions (*Endo et al., 2015*). In fact, ALM has been used as the alternative experimental model for studying limb regeneration (*Nacu et al., 2016*). In this model, a blastema is induced through skin wounding combined with nerve deviation (*Endo et al., 2004*; *Satoh et al., 2007*). When ALM surgery is performed on the anterior side, the induced blastema (ALM blastema) lacks posterior-derived cells and is unable to form a properly patterned limb. This limitation can be overcome by supplying posterior-derived cells via a skin graft from the posterior side, enabling the successful regeneration of a patterned limb.

Similarly, an ALM surgery can be performed on the posterior, dorsal, or ventral side of a limb; in each case, the induced ALM blastema is expected to lack cells from the corresponding opposite orientation (anterior, ventral, or dorsal, respectively). This condition allows the investigation of blastema induction and limb patterning by selectively removing cells from specific orientations of the limb. Analyzing the characteristics and developmental limitations of these ALM blastemas provides valuable insights into the positional identity-mediated signals that regulate axolotl limb regeneration.

The molecular basis of the interpositional interactions within a blastema in the limb patterning process has been partially elucidated (*Nacu et al., 2016*). In a normal blastema, *Fgf8* and *Shh* are expressed at the anterior and the posterior regions, respectively. A previous study demonstrated that FGF8 and SHH proteins can substitute for anterior and posterior tissues, respectively, to form a patterned limb. Moreover, these proteins function as mutually inductive signals, maintaining each other's expression. These findings suggest that FGF8 and SHH serve as the anterior- and posterior-mediated signals, respectively. In contrast, the molecular basis of dorsal- and ventral-mediated signals remains poorly understood. Previous studies have shown that retinoic acid supplementation can convert the dorsal identity to the ventral, enabling patterned limb formation even in the absence of ventral tissues (*McCusker et al., 2014*; *Ludolph et al., 1990*; *Vieira et al., 2019*). However, the ventral-mediated signal remains uncertain, as retinoic acid induces ventral positional identities rather than acting as the ventral-mediated signal itself, suggesting the existence of another, downstream ventral-mediated signal. Consequently, the mechanisms underlying dorsal- and ventral-mediated signals, as well as how they relate to the anterior- and posterior-mediated signals (FGF8 and SHH, respectively), remain unclear.

In the present study, we investigate the roles of dorsal- and ventral-mediated signals in limb patterning. We confirmed the necessity of dorsoventral tissue contact for limb patterning using the ALM. We found that the induction of *Shh* expression was dependent on the co-existence of dorsal and ventral cells in axolotl limb regeneration. Furthermore, we identified WNT10B and FGF2 as dorsal- and ventral-mediated signals, respectively, by RNA-seq analysis and confirmed that these factors mediate *Shh* expression in axolotl blastemas. Our results suggest that the crucial role of the dorsal- and ventral-mediated signals in the limb patterning process is to induce *Shh* expression, thereby enabling anteroposterior interaction. These results contribute to understanding how the integration of four positional identities—dorsal, ventral, anterior, and posterior—drives proper limb patterning during axolotl limb regeneration.

## Results

### Dorsoventral tissue contact is required for limb regeneration in the ALM

We performed the ALM experiment on *A. mexicanum* to directly test whether dorsoventral tissue contact, as well as anteroposterior tissue contact, is required for limb patterning (*Figure 1*). The skin was removed from one side (anterior, posterior, dorsal, or ventral) of a limb, and the large nerve fibers running along the center of a limb (*Nervus medianus* and *Nervus ulnaris*, *Figure 1A*) were dissected and rerouted to the wounded region. We supplied the cells from the contralateral side of a limb as a skin graft in experimental conditions to ensure the presence of sufficient cells for patterned limb formation. We conducted eight types of ALM blastema inductions, which can be categorized into two groups: basic ALM blastemas and skin-grafted ALM blastemas. The basic ALM blastemas included the anteriorly induced ALM blastema (AntBL), the posteriorly induced ALM blastema (PostBL), the dorsally induced ALM blastema (DorBL), and the ventrally induced ALM blastema (VentBL), all of which should lack cells from the contralateral side. The skin-grafted ALM blastemas included AntBL with posterior skin grafting (AntBL + P), PostBL with anterior skin grafting (PostBL + A), DorBL with ventral skin grafting (DorBL + V), and VentBL with dorsal skin grafting (VentBL + D); these ALM blastemas should contain cells with all anteroposterior and dorsoventral orientations. For the ALM experiment, we defined the two large blood vessels running on the anterior and posterior sides at the stylopod level (*Vena cephalica* and *Vena basilica*, *Figure 1A*) as the anterior and posterior dorsoventral borders, respectively. At 10 days post-surgery (dps), an ALM blastema was observed in all experimental groups (*Figure 1B–I*). Consistent with previous studies (*Endo et al., 2004*; *Nacu et al., 2016*), limbs with multiple digits were formed from AntBL + P and PostBL + A (*Table 1*, *Figure 1F, G*), whereas AntBL

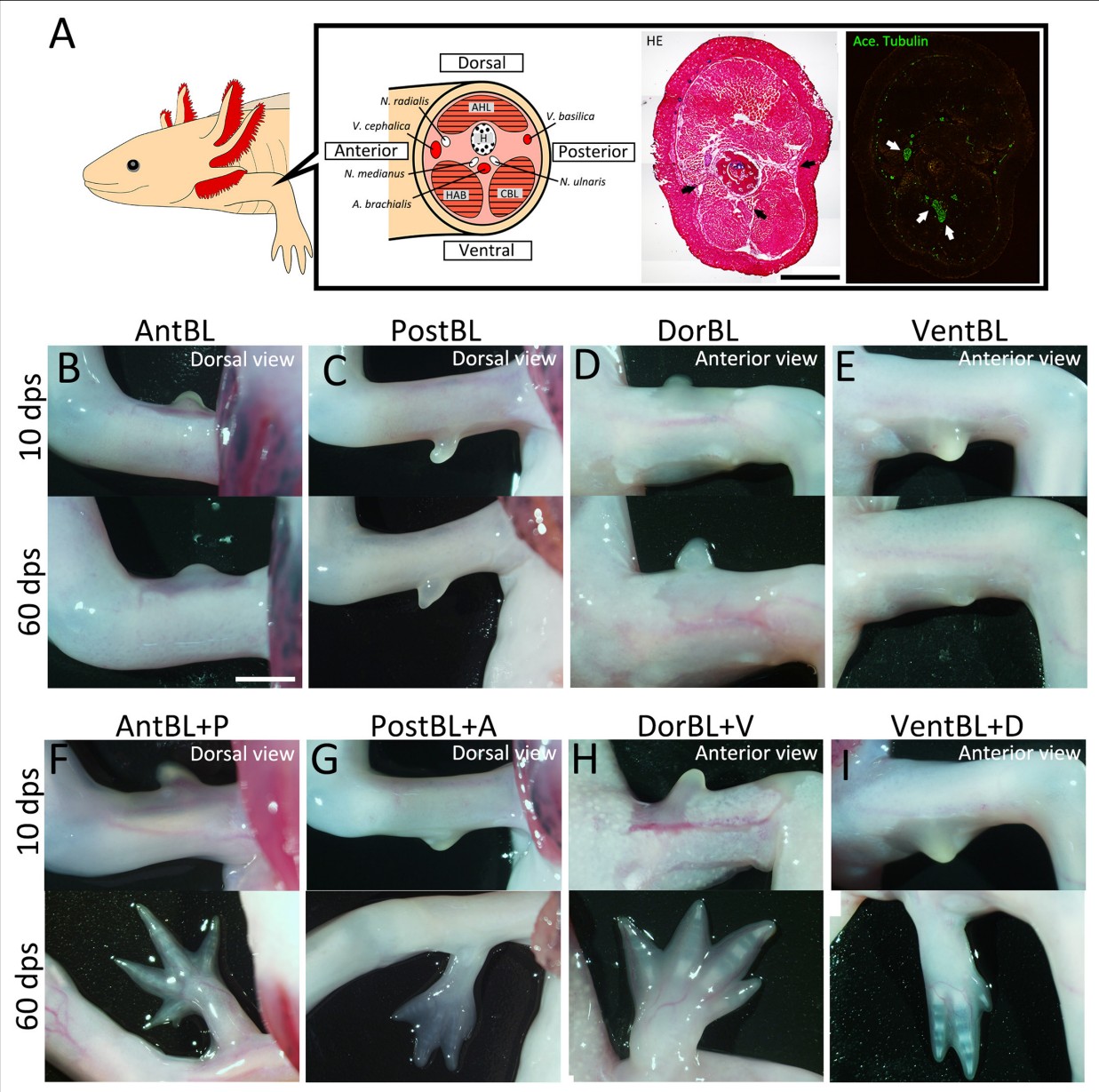

**Figure 1.** Accessory limb model (ALM) experiments at the four orientations. (**A**) Schematic image of anatomy at the stylopod level of axolotl limb. Hematoxylin and eosin (HE) staining (bright field) and acetylated alpha tubulin, visualized by immunofluorescence (green, dark field), are shown in the right panels. Black and white arrows, respectively, indicate major blood vessels and nerves. H: *humerus*, AHL: *Anconaeus humeralis lateralis*, HAB: *Humeroantebrachialis*, CBL: *Coracobrachialis longus*. (**B–I**) Blastemas induced at the anterior, posterior, dorsal, or ventral region by skin wounding plus nerve deviation without (**B–E**) or with (**F–I**) skin grafting from the opposite side of the limb. (F–I) Limb patterning was observed (*n* = 8/9 for F, 4/7 for G, 7/14 for H, and 7/11 for I, see *Table 1* for more detail). Images were captured at 10 and 60 dps. Scale bar = 3 mm (A, B). (B–I) are all shown at the same magnification.

and PostBL either regressed or formed bump structures (*Table 1*, *Figure 1B, C*). Similarly, DorBL + V and VentBL + D formed limbs (*Table 1*, *Figure 1H, I*), whereas most DorBL and VentBL either regressed or formed bump structures (*Table 1*, *Figure 1D, E*). Notably, only 1 out of 12 DorBL formed a limb, which may have resulted from ventral tissue contamination, as the rerouted nerves originally ran along the ventral side of the humerus. In contrast, none of the 22 VentBL formed multi-digit limbs. These results suggest that both dorsal and ventral tissues, as well as anterior and posterior tissues, are required for successful limb formation.

**Table 1.** The induction rate of bump/limb formation in accessory limb model (ALM).

| Experiments | N | Regress | | Bump | | Limb | |
|---|---|---|---|---|---|---|---|
| AntBL | 8 | 3 | (37.5%) | 5 | (62.5%) | 0 | (0.0%) |
| AntBL + P | 9 | 0 | (0.0%) | 1 | (11.1%) | 8 | (88.9%) |
| PostBL | 8 | 6 | (75.0%) | 2 | (25.0%) | 0 | (0.0%) |
| PostBL + A | 7 | 2 | (28.6%) | 1 | (14.3%) | 4 | (57.1%) |
| DorBL | 12 | 8 | (66.7%) | 3 | (25.0%) | 1 | (8.3%) |
| DorBL + V | 14 | 5 | (35.7%) | 2 | (14.3%) | 7 | (50.0%) |
| VentBL | 22 | 15 | (68.2%) | 7 | (31.8%) | 0 | (0.0%) |
| VentBL + D | 11 | 2 | (18.2%) | 2 | (18.2%) | 7 | (63.6%) |

## Gene expression patterns in ALM blastemas

To elucidate the characteristics of the ALM blastemas that typically fail to form limbs (AntBL, PostBL, DorBL, and VentBL), we analyzed gene expression patterns at 10 dps (*Figure 2*, *Figure 2—figure supplement 1*). *Lmx1b*, a gene necessary and sufficient for establishing the dorsal identity in mesenchymal cells in developing limb buds (*Vogel et al., 1995*; *Chen et al., 1998*; *Chen and Johnson, 2002*), was used as a dorsal marker because we previously reported that *Lmx1b* is exclusively expressed in dorsal-derived cells and can be activated in the ALM experiment (*Iwata et al., 2020*; *Yamamoto et al., 2022*). The expression patterns of *Fgf8*, *Shh*, and *Lmx1b* were investigated using in situ hybridization (ISH). In AntBL and PostBL, *Lmx1b* expression was restricted to the dorsal half of the blastemas (*Figure 2B, G*, *Figure 2—figure supplement 1A, D*, n = 5/5 for both), suggesting the presence of both dorsal and ventral tissues. These expression patterns correspond to the anatomically defined dorsoventral borders (*Figure 1A*). In contrast, *Lmx1b* was expressed throughout the entire region of DorBL (*Figure 2L*, *Figure 2—figure supplement 1G*, n = 6/6), whereas no *Lmx1b* expression was detected in VentBL (*Figure 2Q*, *Figure 2—figure supplement 1J*, n = 6/6), suggesting the absence of ventral tissue in DorBL and dorsal tissue in VentBL. Consistent with previous studies, *Fgf8* expression was observed in AntBL (*Figure 2C*, *Figure 2—figure supplement 1B*, n = 4/5), but not in PostBL (*Figure 2H*, *Figure 2—figure supplement 1E*, n = 5/5, *Nacu et al., 2016*). Conversely, *Shh* expression was detected in PostBL (*Figure 2I*, *Figure 2—figure supplement 1F*, n = 5/5), but not in AntBL (*Figure 2D*, *Figure 2—figure supplement 1C*, n = 5/5). In DorBL and VentBL, *Shh* expression was largely absent (*Figure 2N*, *Figure 2—figure supplement 1I*, n = 5/6; *Figure 2S*, *Figure 2—figure supplement 1L*, n = 6/6), whereas *Fgf8* expression was present in both (*Figure 2M*, *Figure 2—figure supplement 1H*, n = 4/6; *Figure 2R*, *Figure 2—figure supplement 1K*, n = 4/6). In DorBL, *Shh* expression was observed in only one out of six samples, possibly due to ventral tissue contamination, as described above. These results suggest that *Fgf8* expression is independent of the co-existence of dorsal and ventral tissues, whereas *Shh* expression appears to require their presence.

## Induction of *Shh* expression requires both dorsal and ventral cells

The absence of *Shh* expression in most DorBL and VentBL, combined with its consistent expression in all PostBL—which contain both *Lmx1b*-positive and *Lmx1b*-negative regions—suggests that the induction of *Shh* expression depends on the presence of both dorsal and ventral cells (*Figure 2*). To test this hypothesis, we performed cell-tracing experiments using GFP-expressing skin from transgenic animals (*Figure 3*). We induced VentBL in leucistic animals and then grafted the posterior half of the dorsal skin (VentBL + PD$_{gfp}$) or the posterior half of the ventral skin (VentBL + PV$_{gfp}$) from GFP transgenic animals (*Figure 3A*). Similarly, DorBL was induced in wild-type animals, and the posterior half of the dorsal skin (DorBL + PD$_{gfp}$) or the posterior half of the ventral skin (DorBL + PV$_{gfp}$) from GFP transgenic animals was grafted (*Figure 3A*). GFP-positive mesenchymal cells derived from the posterior skin were expected to express *Shh* if provided with a suitable environment, because posteriorly derived cells, not anteriorly derived cells, are known to have the competency to express *Shh* in a blastema—that is, whether a cell is capable of expressing *Shh* depends on its original positional identity (*Iwata et al., 2020*), but whether it actually expresses *Shh* should depend on the environment

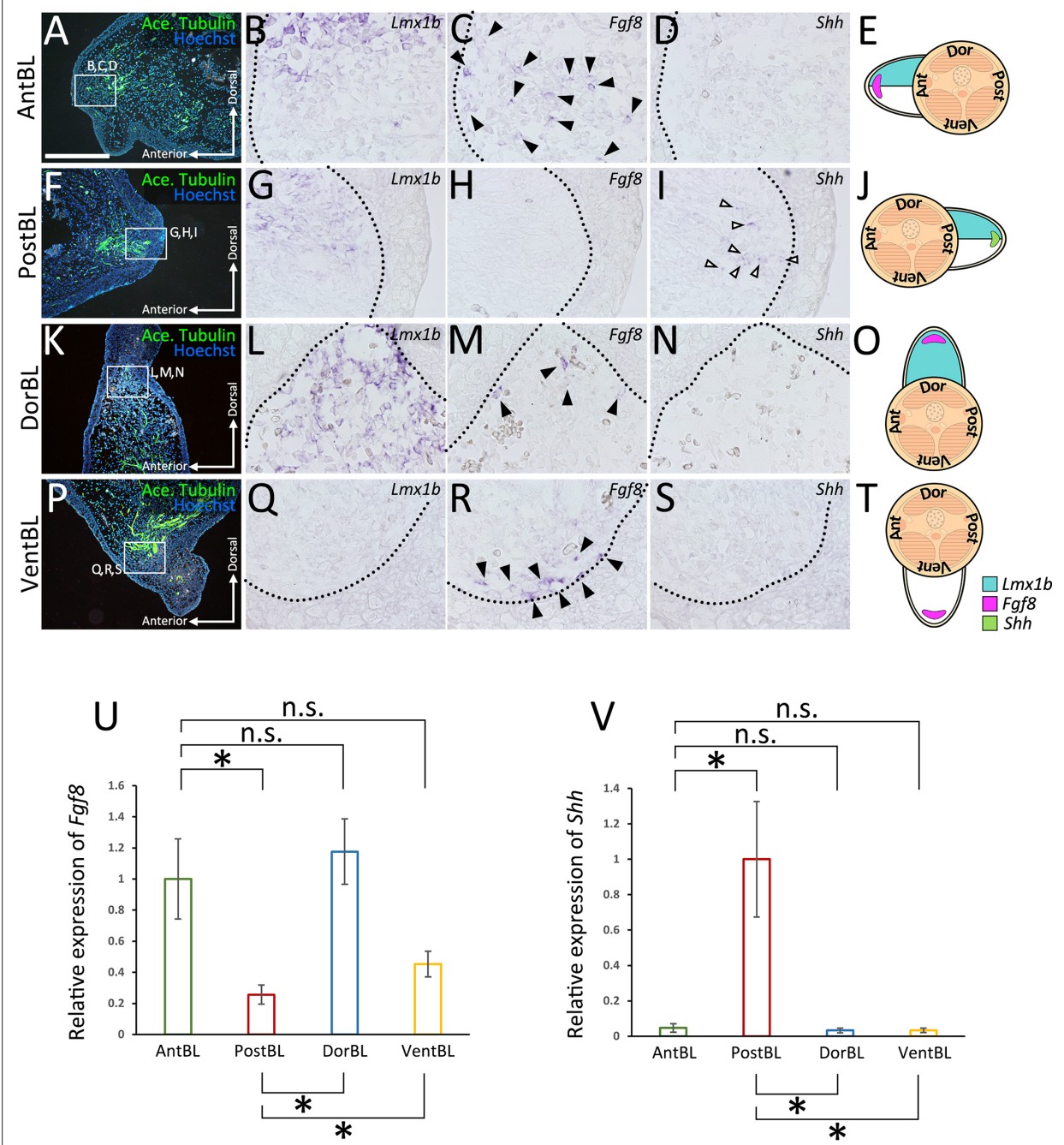

**Figure 2.** Gene expression patterns of the accessory limb model (ALM)-induced blastemas. Sections of anteriorly (**A–D**), posteriorly (**F–I**), dorsally (**K–N**), or ventrally (**P–S**) induced blastemas at 10 dps. Acetylated alpha tubulin (**A–P**) was visualized by immunofluorescence. Expression of *Lmx1b* (**B–Q**), *Fgf8* (**C–R**), and *Shh* (**D–S**) in the regions indicated by white boxes in (**A–P**) was visualized by in situ hybridization. Images of the entire blastema are provided in *Figure 2—figure supplement 1*. *Fgf8* expression was observed in AntBL (**C**), DorBL (**M**), and VentBL (**R**) (n = 4/5, 4/6, and 4/6, respectively). *Shh* expression was observed in PostBL (**I**) (n = 5/5). In each case, these expression patterns of *Fgf8* and *Shh* were focal and only a few cells expressed *Fgf8* or *Shh*. Black and white arrowheads indicate the signals of *Fgf8* and *Shh* expression, respectively. The dotted line indicates the epithelial–mesenchyme border. (**E–T**) Schematic images of gene expression patterns. (**U, V**) Quantitative analysis of *Fgf8* and *Shh* expression in ALM blastemas. Data are presented as mean ± SEM (n = 5 for all groups). n.s.: no significant difference, *p < 0.05, **p < 0.005 (two-tailed Welch's *t*-test). Scale bar in (**A**) = 700 μm. (**A–P**) are all shown at the same magnification.

The online version of this article includes the following figure supplement(s) for figure 2:

**Figure supplement 1.** Gene expression patterns across the entire accessory limb model (ALM)-induced blastemas.

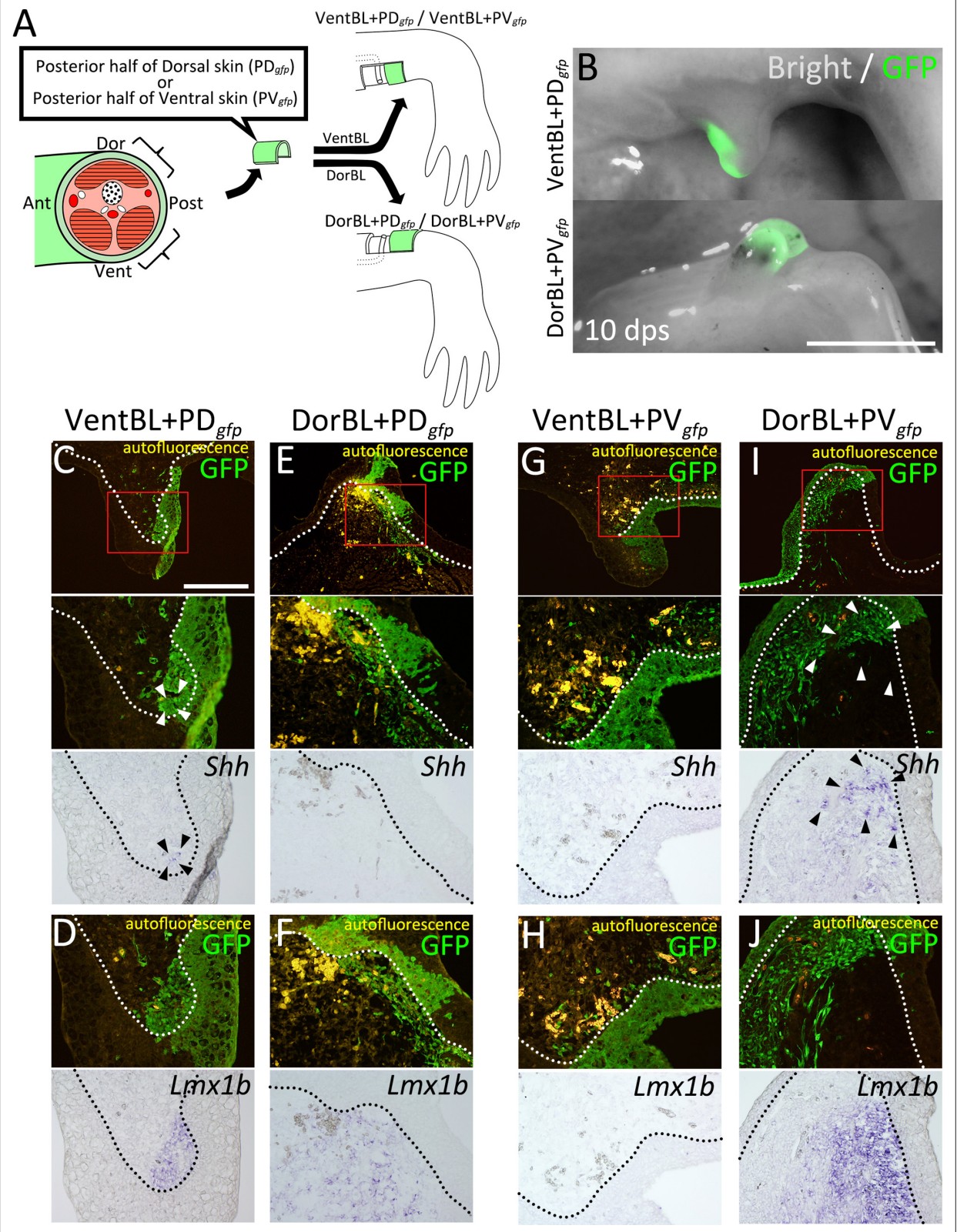

**Figure 3.** Co-existence of dorsal and ventral cells induces Shh expression. (**A**) Experimental scheme. Posterior half of dorsal (PD$_{gfp}$) or ventral (PV$_{gfp}$) GFP-expressing skin was grafted on VentBL (VentBL + PD$_{gfp}$; **C, D**/VentBL + PV$_{gfp}$; **G, H**), or DorBL (DorBL + PD$_{gfp}$; **E, F**/DorBL + PV$_{gfp}$; **I, J**) region. (**B**) Induced blastemas at 10 dps; images of bright and dark fields are merged. (**C–J**) Dark and bright fields of the same sections of induced blastemas at 10 dps. Red boxes in (**C–I**) indicate the corresponding regions of lower images. Expression of *Shh* and *Lmx1b* was visualized by in situ hybridization. GFP signals

*Figure 3 continued on next page*

*Figure 3 continued*

were visualized by immunofluorescence. Arrowheads indicate the cells expressing *Shh*. *Shh* expression was observed in VentBL + PD$_{gfp}$ (**C**) and DorBL + PV$_{gfp}$ (**I**) (*n* = 4/6 and 3/6, respectively), but not in DorBL + PD$_{gfp}$ (**E**) and VentBL + PV$_{gfp}$ (**G**) (*n* = 6/6 and 7/7, respectively). The dotted line indicates the epithelial–mesenchyme border. For all samples, we collected serial sections spanning the entire blastema. For blastemas in which *Shh* expression was observed, we present representative sections showing the signal. For blastemas without detectable *Shh* expression, we present a section from the central region that contains GFP-positive cells. Scale bar = 3 mm (**B**) and 700 μm (**C**). (**C–J**) are all shown at the same magnification.

in which the cell is placed. If the induction of *Shh* expression depends on the co-existence of dorsal and ventral cells, *Shh* expression in the GFP-positive cells should be observed in VentBL + PD$_{gfp}$ and DorBL + PV$_{gfp}$ but not in DorBL + PD$_{gfp}$ or VentBL + PV$_{gfp}$. Samples were collected at 10 dps, and the expression of *Shh* and *Lmx1b* was analyzed using ISH. In all four experimental groups, *Lmx1b* expression patterns in the GFP-negative regions were consistent with those in DorBL and VentBL (*Figure 3D, F, H, J*). In DorBL + PD$_{gfp}$ and VentBL + PD$_{gfp}$, *Lmx1b* expression was observed in GFP-positive mesenchymal cells, whereas GFP-positive cells in DorBL + PV$_{gfp}$ and VentBL + PV$_{gfp}$ were *Lmx1b*-negative (*Figure 3C–F*). These *Lmx1b* expression patterns suggest that dorsoventral tissue contact was present in VentBL + PD$_{gfp}$ and DorBL + PV$_{gfp}$ but absent in DorBL + PD$_{gfp}$ and VentBL + PV$_{gfp}$. In the GFP-positive cells derived from PD$_{gfp}$ skin, *Shh* expression was observed in VentBL + PD$_{gfp}$ (*Figure 3C*, *n* = 4/6) but not in DorBL + PD$_{gfp}$ (*Figure 3E*, *n*=6/6). Similarly, in the GFP-positive cells derived from PV$_{gfp}$ skin, *Shh* expression was observed in DorBL + PV$_{gfp}$ (*Figure 3I*, *n* = 3/6) but not in VentBL + PV$_{gfp}$ (*Figure 3G*, *n* = 7/7). In addition, *Shh* expression was observed in some GFP-negative mesenchymal cells in samples where *Shh* expression was observed in GFP-positive cells. These results suggest that the induction of *Shh* expression in posteriorly derived cells requires the co-existence of dorsal and ventral cells.

## Limb formation in the absence of dorsoventral tissue contact

DorBL and VentBL failed to form limbs (*Figure 1D, E*). *Shh* expression was absent in these blastemas, whereas *Fgf8* was expressed (*Figure 2*). Results from the cell-tracing experiments suggest that *Shh* expression requires the co-existence of dorsal and ventral cells (*Figure 3*). Based on these findings, we hypothesized that while the co-existence of dorsal and ventral cells is necessary to induce *Shh* expression, it is not directly required for limb patterning if SHH protein is externally provided. To test this hypothesis, we overexpressed *Shh* in DorBL and VentBL (*Figure 4A, B*). As a positive control, *Shh* was overexpressed in AntBL, where *Fgf8* was expressed but *Shh* expression was absent (*Figure 2C, D*), because a previous study has shown that *Shh* overexpression to AntBL can induce limb patterning (*Nacu et al., 2016*). As a result, *Shh*-electroporated AntBL, DorBL, and VentBL formed limbs (*n* = 10/19, 6/18, and 8/14, respectively, *Figure 4C–G*). In contrast, GFP-electroporated DorBL and VentBL (negative controls) failed to form limbs (*n* = 7/7 and 6/6, respectively). We further analyzed the anatomy of the induced limbs (*Figure 4H–K*, *Figure 4—figure supplement 1*). Because ALM-induced limbs frequently exhibit abnormal and highly variable morphologies, which makes it difficult to use consistent anatomical landmarks such as particular digits or muscle groups, we focused our analysis on morphological symmetry. We found that limbs formed from DorBL and VentBL exhibited symmetric structures along the dorsoventral axis, suggesting double-dorsal and double-ventral structures, respectively (*Figure 4J, K*, *Figure 4—figure supplement 1C, D*). In contrast, limbs formed from AntBL, which contained both *Lmx1b*-positive and *Lmx1b*-negative regions (*Figure 2B*), exhibited asymmetric structures similar to those of normal limbs (*Figure 4H, I*, *Figure 4—figure supplement 1A, B*). To evaluate the symmetry of the limbs, we applied a machine learning-based method using ilastik because staining intensity varied among samples, such that a region identified as 'muscle' in one sample could be assigned differently in another if classification were based solely on color. The machine-learning classifier trained separately for each sample allowed us to group the same tissues consistently within that sample irrespective of intensity differences. To minimize the effects of curvature or fixation-induced distortion, the boxes with a width of 400 μm were selected, and the angle was adjusted so that the outer contour (epidermal surface) was aligned symmetrically; this procedure was applied uniformly across all conditions to avoid bias. Each pixel in the images was classified into five classes (Classes 1–5, *Figure 4L*). In this process, we annotated regions of background (Class 1), cartilage (Class 2), muscle (Class 3), other connective tissue (Class 4), and epidermis (Class 5) as training data for classification. Using these annotated regions as references, pixels were automatically classified into

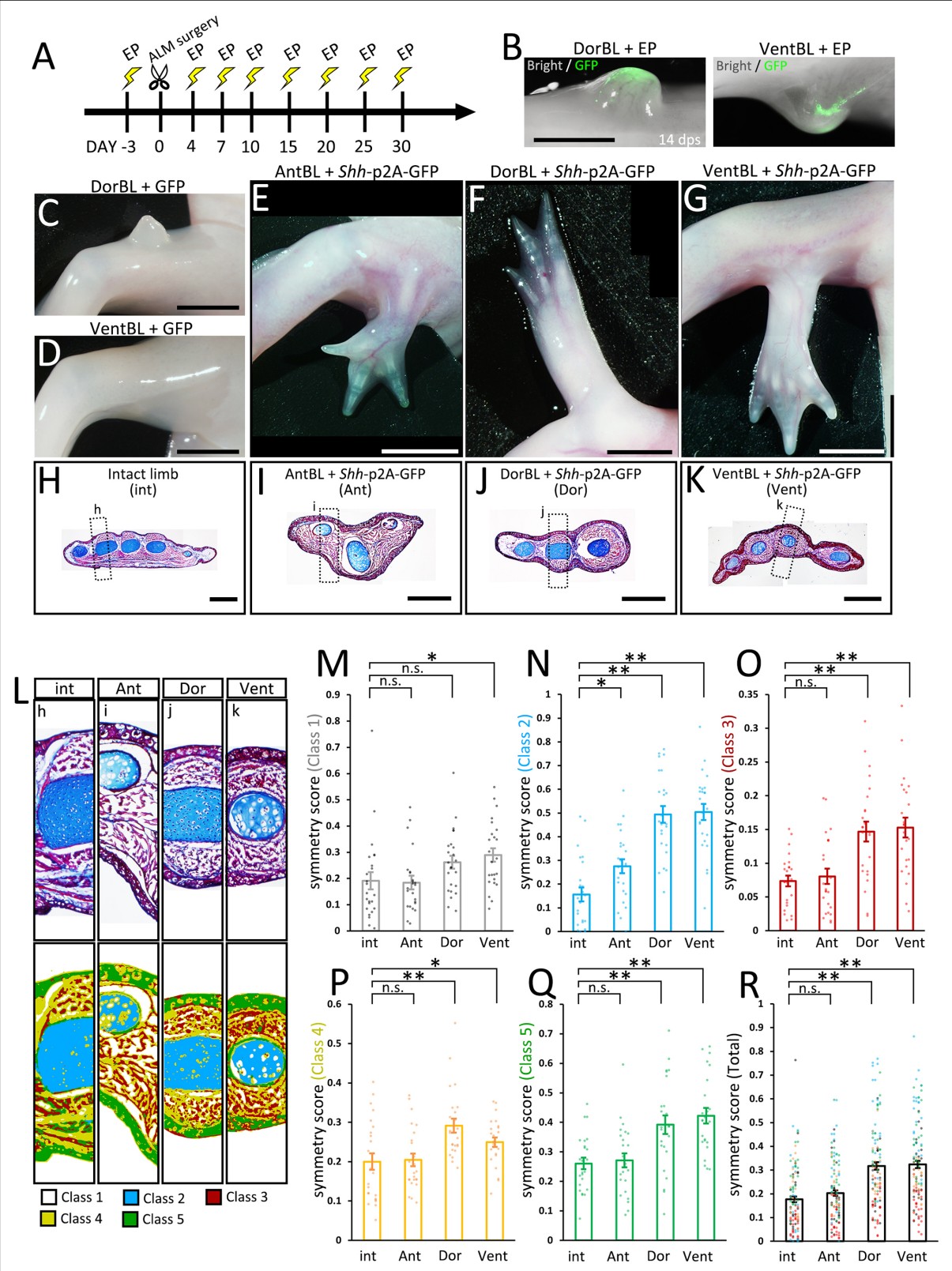

**Figure 4.** Limb formation without the co-existence of dorsal and ventral cells by Shh overexpression. (**A**) Experimental scheme. *Shh*-p2A-GFP- or GFP-containing pCS2 vector was electroporated (EP) into AntBL, DorBL, and VentBL. (**B**) Induced blastemas at 14 dps; images of bright and dark fields are merged. (**C–G**) Phenotypes at 90 dps. Limb patterning was observed in (**E–G**) (*n* = 10/19, 6/18, and 8/14, respectively). (**H–K**) Histological analysis of intact and induced limbs. Standard Masson's trichrome staining was performed on the transverse sections. The dotted boxes indicate the regions shown

*Figure 4 continued on next page*

*Figure 4 continued*

in (**L**). (**L**) Upper panels: analyzed regions for calculating symmetry scores. Lower panels: images after pixel classification by machine learning. (**M–R**) Symmetry scores of each class. Scores obtained from the same limb are plotted at the same *x*-coordinates. Data are presented as mean ± SEM. n.s.: no significant difference, *$p < 0.05$, **$p < 0.005$ (two-tailed Welch's *t*-test). Scale bar = 2 mm (**B**), 4 mm (**C–G**), and 1 mm (**H–K**).

The online version of this article includes the following source code and figure supplement(s) for figure 4:

**Source code 1.** Source code for calculating the symmetry score.

**Figure supplement 1.** Histological analysis of intact limbs and limbs induced by Shh electroporation.

the five classes. Each class was assumed to primarily represent these tissues and regions with similar characteristics. Symmetry scores were then calculated for each class individually and for the combined set of all classes (*Figure 4M–R*, see Materials and methods). The analysis revealed that symmetry scores for Classes 2 and 3, which encompass cartilage and muscle, and scores for the combined set of all classes were significantly higher in limbs formed from DorBL and VentBL compared to intact limbs (*Figure 4N, O, R*). In contrast, the differences in symmetry scores for Classes 1, 4, and 5 in these limbs were relatively small (*Figure 4M, P, Q*), likely because the axis of symmetry was manually set to maximize the external shape as symmetrically as possible. No significant differences in symmetry scores across all classes were observed between intact limbs and limbs formed from AntBL. These results suggest that limbs formed from DorBL and VentBL exhibit symmetric internal structures compared to normal limbs. Therefore, *Shh* overexpression appears to compensate for the lack of co-existence of dorsal and ventral cells in limb patterning, without inducing new dorsal or ventral identities. Thus, we conclude that the co-existence of both dorsal and ventral cells is critical for inducing *Shh* expression, which in turn is essential for limb patterning.

## The molecular basis of the dorsal- and ventral-mediated signals

To investigate the molecular basis of dorsal- and ventral-mediated signals, we performed RNA-seq analysis on DorBL and VentBL and identified differentially expressed genes (DEGs) between the two groups ($p < 0.05$, *Figure 5A*). Among the DEGs, we specifically focused on genes annotated as 'intercellular signaling molecules' in PANTHER and identified 21 genes (*Figure 5B*). In this analysis, we found that 5 genes were expressed at higher levels in DorBL and 16 genes were expressed at higher levels in VentBL (*Figure 5B*). Notably, *Wnt4*, *Wnt10b*, *Fgf2*, *Fgf7*, and *Tgfb2*, which encode secreted proteins, belong to the WNT, FGF, and TGFB families, which play key roles in major signaling pathways regulating limb developmental processes, and we focused on these five genes. To examine whether these genes regulate the induction of *Shh* expression in ALM blastemas, we overexpressed each gene in either DorBL or VentBL. As a result, *Shh* expression was observed in *Wnt10b*-electroporated VentBL ($n = 4/5$) and *Fgf2*-electroporated DorBL ($n = 5/7$, *Figure 5C*). In contrast, *Shh* expression was not detected in *Wnt10b*-electroporated DorBL ($n = 6/6$) or *Fgf2*-electroporated VentBL ($n = 5/5$). Similarly, *Shh* expression was not detected in *Fgf7*- or *Tgfb2*-electroporated DorBL ($n = 5/5$ for both) or in *Wnt4*-electroporated VentBL ($n = 6/6$). To confirm the ISH data, we performed RT-qPCR on *Fgf2*- or GFP (control)-electroporated DorBL and *Wnt10b*- or GFP (control)-electroporated VentBL and observed significant upregulation in *Shh* expression (*Figure 5D, E*). In *Wnt10b*-electroporated VentBL, *Axin2*, a downstream transcriptional target of canonical WNT signaling, and *Lef1*, a canonical WNT pathway effector expressed in axolotl limb mesenchyme (*Glotzer et al., 2022*), were also upregulated (*Figure 5F, G*). Next, to confirm *Shh* induction by WNT signaling in VentBL, we treated VentBL with 1 µM 6-bromoindirubin-3-oxime (GSK-3 Inhibitor, BIO). As a result, *Shh* expression was relatively higher in BIO-treated VentBL compared to DMSO-treated control VentBL (*Figure 5—figure supplement 1B, D, E*). Additionally, symmetric limbs were formed in *Wnt10b*-electroporated VentBL ($n = 3/12$), BIO-treated VentBL ($n = 3/8$), and in *Fgf2*-electroporated DorBL ($n = 5/10$), consistent with the results of *Shh* overexpression (*Figure 5H–J*, *Figure 5—figure supplements 1C, F–H and 2A, B*). These findings suggest that WNT10B, expressed highly in dorsal blastema cells, and FGF2, expressed highly in ventral blastema cells, function as the dorsal- and ventral-mediated signals, respectively, to induce *Shh* expression, which subsequently supports limb patterning.

To test whether our model applies to normal regeneration, we analyzed *Wnt10b* and *Fgf2* expression in amputation-induced blastemas (*Figure 5—figure supplement 3*). We first performed ISH on blastemas at several stages (early bud [EB], middle bud [MB], and late bud [LB]), but the signals were

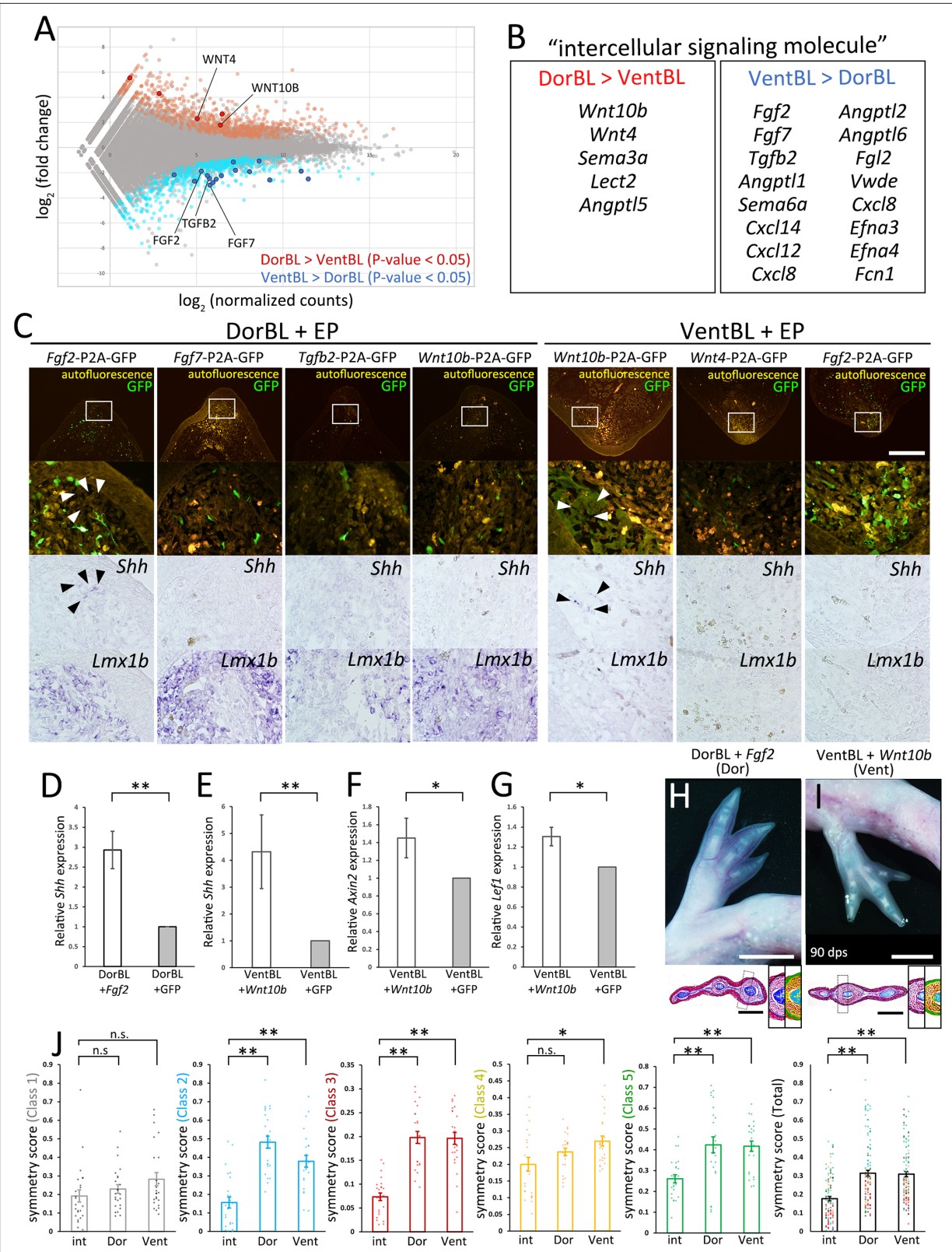

**Figure 5.** Identification of candidate molecules of the dorsal- and ventral-mediated signals. RNA-seq was performed on DorBL and VentBL at 10 dps. (**A**) MA plot of the result. (**B**) List of differentially expressed genes (DEGs) annotated as 'intercellular signaling molecules'. (**C**) Bright and dark fields of sections of DorBL and VentBL at 10 dps with candidate genes introduced. Expression of *Shh* and *Lmx1b* was visualized by in situ hybridization, and GFP signals were visualized by immunofluorescence. Arrowheads indicate the cells expressing *Shh*. The white boxes indicate the regions of the lower panels.

*Figure 5 continued on next page*

*Figure 5 continued*

Images of dark and bright fields of *Shh* are obtained from the same section. For samples with detectable *Shh* expression, the window was placed in the region where the signal was observed, and for conditions without detectable *Shh* expression, the window was positioned in a comparable region containing GFP-positive cells. *Shh* expression was observed in *Wnt10b*-electroporated VentBL ($n = 4/5$) and *Fgf2*-electroporated DorBL ($n = 5/7$), but not in *Wnt10b*-electroporated DorBL ($n = 6/6$), *Fgf2*-electroporated VentBL ($n = 5/5$), *Fgf7*-electroporated DorBL ($n = 5/5$), *Tgfb2*-electroporated DorBL ($n = 5/5$), or in *Wnt4*-electroporated VentBL ($n = 6/6$). (D, E) Quantitative analysis of *Shh* expression in *Fgf2*- or GFP-electroporated DorBL and *Wnt10b*- or GFP-electroporated VentBL. Data are presented as mean ± SEM ($n = 7$ for both). In each case, *Fgf2* or *Wnt10b* was electroporated into the DorBL or VentBL induced in the left limb, and GFP was electroporated into the contralateral right limb of the same animal. (F, G) Quantitative analysis of *Axin2* and *Lef1* expression in *Wnt10b*- or GFP-electroporated VentBL. Data are presented as mean ± SEM ($n = 7$ for both). (H, I) The limbs formed from VentBL with *Wnt10b* and DorBL with *Fgf2* at 90 dps. Histological analysis and pixel classification were performed in the same way as in *Figure 4*. The dotted boxes indicate the regions shown in the right panels. (J) Symmetry scores of each class. Scores obtained from the same limb are plotted at the same *x*-coordinates. The plots of 'int' are the same plots as in *Figure 4*. Data are presented as mean ± SEM. n.s.: no significant difference, *p<0.05, **p<0.005 (two-tailed paired *t*-test for D–G, and two-tailed Welch's *t*-test for J). Scale bar = 700 µm (C), 4 mm (H, I, upper panels), and 1 mm (H, I, lower panels). Images in (C) are all shown at the same magnification.

The online version of this article includes the following figure supplement(s) for figure 5:

**Figure supplement 1.** Limb patterning from BIO-treated VentBL.

**Figure supplement 2.** Histological analysis of limbs induced by Fgf2 or Wnt10b electroporation.

**Figure supplement 3.** Gene expression patterns during normal limb regeneration.

**Figure supplement 4.** Gene expression patterns in a normal blastema assessed by reanalysis of axolotl single-cell RNA-seq data.

weak and inconsistent, and we could not reliably detect clear expression domains (*Figure 5—figure supplement 3A*). We then performed RT-qPCR on manually microdissected dorsal and ventral halves of MB blastemas (*Figure 5—figure supplement 3B*). We found that *Wnt10b* and *Fgf2* were expressed at significantly higher levels in the dorsal and ventral halves, respectively, compared to the opposite half. This dorsoventral-biased expression of *Wnt10b* and *Fgf2* is consistent with our RNA-seq data from ALM blastemas. We next quantified *Wnt10b*, *Fgf2*, and *Shh* expression across stages (intact, EB, MB, LB, and early digit [ED]) and found that *Wnt10b* and *Fgf2* expression peaked at the MB stage, whereas *Shh* expression peaked later, at the LB stage (*Figure 5—figure supplement 3C*). This temporal offset in *Shh* upregulation relative to *Wnt10b* and *Fgf2* supports a model in which WNT10B and FGF2 act upstream to induce *Shh* expression.

To identify the cell populations expressing *Wnt10b* and *Fgf2* during normal regeneration, we reanalyzed published single-cell RNA-seq data from a 7 dpa (MB) blastema (*Li et al., 2021*, *Figure 5—figure supplement 4*). The dataset was reclustered, and clusters were assigned using known markers (*Prrx1* for mesenchyme and *Krt17* for epithelium, *Figure 5—figure supplement 4A, B*). As expected, *Lmx1b*, *Fgf8*, and *Shh* were detected in the mesenchymal cluster (*Figure 5—figure supplement 4C, F, G*). *Fgf2* was also expressed in the mesenchymal cluster (*Figure 5—figure supplement 4E*). In contrast, *Wnt10b* expression was detected in both mesenchymal and epithelial clusters (*Figure 5—figure supplement 4D*), but these results may partially reflect technical bias, as low-level signals of epithelial and CT/fibroblast markers can be detected outside their expected clusters (*Figure 5—figure supplement 4A, B*). Both *Wnt10b* and *Fgf2* were expressed in only a few cells, consistent with the ISH data (*Figure 5—figure supplement 3A*). We then examined the relationships between these genes. *Fgf8* and *Shh* were expressed in both *Lmx1b*-positive and *Lmx1b*-negative cells (*Figure 5—figure supplement 4H, I*), but *Fgf8* and *Shh* themselves were mutually exclusive (*Figure 5—figure supplement 4M*). These expression patterns of *Fgf8*, *Shh*, and *Lmx1b* in the normal blastema are consistent with those observed in ALM blastemas (*Figure 2*). For *Wnt10b* and *Fgf2*, their expression did not follow *Lmx1b* expression (*Figure 5—figure supplement 4J, K*), and *Wnt10b* and *Fgf2* themselves were not exclusive (*Figure 5—figure supplement 4L*). Together with the RT-qPCR data (*Figure 5—figure supplement 3B*), these results suggest that *Wnt10b* and *Fgf2* are not exclusively confined to purely dorsal or ventral cells at the single-cell level, even though they show dorsoventral bias when assessed in bulk tissue.

## Limb formation without nerve deviation and ventral skin grafting

A previous study demonstrated that a cocktail of BMP2, FGF2, and FGF8 can substitute for nerve deviation in the ALM experiment on the anterior region, suggesting that nerves contribute to blastema induction by supplying these proteins (*Makanae et al., 2014a*). We found that FGF2 can also

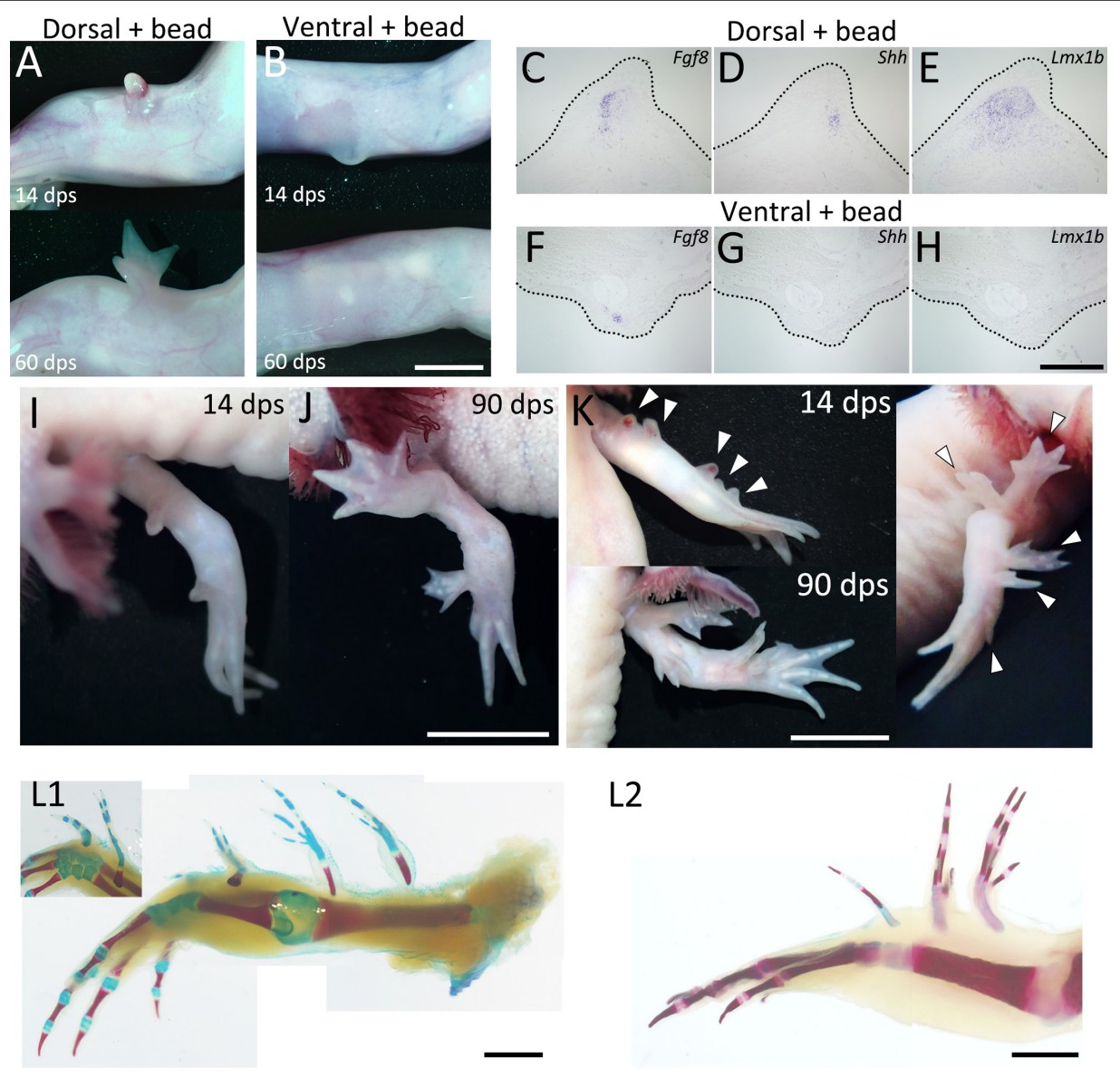

**Figure 6.** Limb formation at the dorsal region by BMP2 + FGF2 + FGF8 supplementation. 14 and 60 dps phenotypes of BMP2 + FGF2 + FGF8 supplementation by bead grafting at the dorsal (**A**) or ventral (**B**) region. Neither nerve deviation nor skin grafting was performed. Limb patterning was observed in the dorsa group (**A**) (n = 12/20) but not in the ventral group (**B**) (n = 0/17). (**C–H**) Expression patterns of *Fgf8*, *Shh*, and *Lmx1b* of induced blastemas at 10 dps. Gene expression was visualized by in situ hybridization. The dotted line indicates the external shape of the blastema. *Fgf8* expression was detected in both dorsal and ventral groups (n = 8/8 for C and 5/5 for **F**), whereas *Shh* expression was detected only in the dorsal group (**D**) (n = 8/8) and not in the ventral group (**G**) (n = 0/5). (**I, J**) Phenotype obtained by BMP2 + FGF2 + FGF8 supplementation to two dorsal regions of an identical limb. (**K, L**) Phenotype obtained by BMP2 + FGF2 + FGF8 supplementation to five dorsal regions of an identical limb (n = 16/35). Scale bar = 3 mm (**B**), 700 µm (**H**), 2 cm (**J**), and 1 cm (**K, L**). (A, B), (C–H), or (I, J) are shown at the same magnification.

substitute for ventral tissue in DorBL (*Figure 5*), suggesting that FGF2 serves not only as part of the nerve factors in blastema induction but also as the ventral-mediated signal in limb patterning. To investigate whether supplementation with BMP2, FGF2, and FGF8 to the dorsal region could substitute for both nerve deviation and ventral skin grafting, we performed experiments involving gelatin beads soaked in these proteins (*Figure 6*). The dorsal or ventral skin was removed at the zeugopod level, and a BMP2 + FGF2 + FGF8-soaked gelatin bead was grafted at 3 dps without nerve deviation. Blastema induction was observed in both experimental groups (*Figure 6A, B*, upper panels). We investigated gene expression patterns in the induced blastemas and found that both *Fgf8* and *Shh* were expressed in blastemas induced at the dorsal site (n = 8/8), whereas *Shh* expression was

absent in ventral blastemas (*n* = 5/5), although *Fgf8* expression was detected in most cases (*n* = 4/5, *Figure 6C, D, F, G*). *Lmx1b* expression patterns corresponded to those observed in DorBL and VentBL (*Figure 6E, H*, *n* = 8/8 and 5/5, respectively). Furthermore, limb patterning was observed in the dorsal groups (*n* = 12/20) but not in the ventral groups (*n* = 17/17, *Figure 6A, B*, lower panels). These findings demonstrate that a straightforward procedure involving dorsal skin wounding and supplementation with BMP2, FGF2, and FGF8 is sufficient to induce accessory limb formation. This method enabled the induction of an ectopic limb without surgical nerve deviation, but a previous study has shown that fine nerve ingrowth can still occur when blastemas are induced by a BMP2 + FGF2 + FGF8-soaked bead (*Makanae et al., 2014b*). These recruited nerves may be functional after blastema induction. Remarkably, this method also enabled the induction of multiple limbs within the same limb. Grafting BMP2 + FGF2 + FGF8-soaked beads to multiple dorsal sites resulted in the formation of multiple limbs along the proximodistal axis (*n* = 16/35, *Figure 6I–L*). Despite careful implantation designed to avoid injuring deep tissues, one sample displayed a fusion of the stylopod with the host humerus—a phenotype associated with deep wounding (*Satoh et al., 2010*; *Makanae et al., 2014a*). This suggests that contributions from a broader cellular population cannot be excluded. However, because this fusion was not consistently observed, and because ectopic limbs induced at the forearm (zeugopod) level did not exhibit such fusion (*n* = 1/6 for stylopod-level inductions; *n* = 0/10 for zeugopod-level inductions, *Figure 6L1, L2*), the data suggest that most ectopic limbs would have developed without substantial ventral-cell contribution. These results refine our understanding of the role of FGF2 in the ALM, particularly its critical function on the dorsal side.

## Discussion

### Dorsal- and ventral-mediated signals are required for the induction of Shh expression

In axolotl limb regeneration, cells derived from all four positional origins—anterior, posterior, dorsal, and ventral—are required for an induced blastema to form a limb. This was confirmed through the ALM experiments (*Figure 1*). The gene expression patterns in these ALM blastemas revealed distinct molecular characteristics for each group (*Figure 2*). *Fgf8* expression in AntBL and *Shh* expression in PostBL are consistent with a previous report (*Nacu et al., 2016*). The absence of SHH in AntBL and of FGF8 in PostBL likely accounts for their failure to form limbs. *Lmx1b* expression in the dorsal half of both AntBL and PostBL suggests that both dorsal and ventral tissues co-existed in these blastemas (*Figure 2B, G*, *Figure 2—figure supplement 1A, D*). In contrast, DorBL and VentBL exhibited distinct *Lmx1b* expression patterns, suggesting that most cells in DorBL and VentBL are derived from dorsal or ventral origins, respectively (*Figure 2L, Q*, *Figure 2—figure supplement 1G, J*). A possible explanation for the lack of *Shh* expression in DorBL and VentBL is that the induction of *Shh* expression may depend on the co-existence of dorsal and ventral cells, although *Fgf8* can be expressed independently of the co-existence of dorsal and ventral cells (*Figure 2M, N, R, S, U, V*, *Figure 2—figure supplement 1H, I, K, L*). This idea is supported by the cell-tracing experiment (*Figure 3*). In these assays, the posteriorly derived cells expressed *Shh* only when both dorsal and ventral cells were present. These results strongly suggest that the co-existence of dorsal and ventral cells is necessary for inducing *Shh* expression. Consequently, one of the essential roles of the dorsal and ventral cells in limb patterning is to mediate signals required for *Shh* induction in the posterior cells, which facilitates the anteroposterior interaction.

Our findings clarify the hierarchical relationship between dorsal- and ventral-mediated signals and anteroposterior interaction. Our data demonstrate that dorsal and ventral cells are essential for inducing *Shh* expression in a regenerating blastema. Furthermore, we showed that DorBL and VentBL, which typically fail to form a limb, could form patterned limbs with ectopic *Shh* expression, even in the absence of dorsal and ventral cell co-existence (*Figure 4*). Furthermore, such limbs exhibited dorsally or ventrally symmetric structures (*Figure 4F, G, J–R*, *Figure 4—figure supplement 1C, D*), highlighting that dorsoventral patterning depends on the cell origin. Thus, we concluded that dorsal- and ventral-mediated signals are essential for patterned limb formation via inducing *Shh* expression.

## WNT10B and FGF2 as dorsal- and ventral-mediated signals

We identified *Wnt10b* and *Fgf2* as candidate genes encoding the dorsal- and ventral-mediated signals required to induce *Shh* expression, respectively (*Figure 5*). Our DEG analysis between DorBL and VentBL revealed higher *Wnt10b* expression in DorBL and higher *Fgf2* expression in VentBL (*Figure 5A, B*). *Wnt10b*-electroporation in VentBL and *Fgf2*-electroporation in DorBL induced *Shh* expression and subsequent limb patterning (*Figure 5C–E*). *Wnt4* expression was also elevated in DorBL, while *Fgf7* and *Tgfb2* were upregulated in VentBL. However, we did not observe *Shh* induction in the *Fgf7*-electroporated or *Tgfb2*-electroporated DorBL or in the *Wnt4*-electroporated VentBL. FGF7 is known to be capable of inducing the apical ectodermal ridge (AER) in chick limb development (*Yonei-Tamura et al., 1999*), but little is known about its role in limb regeneration. In newt limb regeneration, KGFR, which acts as an FGF7 receptor, is expressed in the basal layer of the wound epithelium, while FGFR1, which acts as an FGF2 receptor, is primarily expressed in the mesenchyme (*Poulin et al., 1993*). This differential expression pattern suggests that FGF2 and FGF7 target distinct cell populations, potentially explaining the differences in their ability to induce *Shh* expression. The difference in the ability of *Wnt10b* and *Wnt4* to induce *Shh* expression in VentBL may reflect differences in how these ligands activate downstream WNT signaling programs. WNT10B is a potent activator of the canonical WNT signaling pathway (*Bennett et al., 2005*), although WNT10B has also been reported to trigger a β-catenin-independent pathway (*Lin et al., 2021*). Similarly, WNT4 can activate both the canonical WNT signaling pathways and a non-canonical, β-catenin-independent pathway (*Li et al., 2013*; *Li et al., 2019*). We also observed *Shh* induction in BIO-treated VentBL, indicating that the canonical WNT signaling pathway regulates *Shh* expression. However, it is uncertain why *Shh* expression was observed in *Wnt10b*-electroporated VentBL, but not in *Wnt4*-electroporated VentBL. One possible explanation is that different WNT ligands can engage the same receptors (Frizzled/LRP6) yet elicit distinct downstream programs, suggesting that such ligand-specific outputs may vary depending on cell context (*Voss et al., 2025*). Regarding *Tgfb2*, TGF-β signaling has been implicated in blastema induction during salamander limb regeneration (*Sader and Roy, 2022*; *Lévesque et al., 2007*). However, little is known about the relationship between TGF-β signaling and *Shh* induction in limb development and regeneration, suggesting that TGF-β signaling does not play a primary role in *Shh* induction. Consistent with this, we did not observe *Shh* induction in the *Tgfb2*-electroporated DorBL, as described above. This suggests that TGFB2 signaling may be involved in dorsoventral regulation independent of *Shh* induction. Taken together, among the candidate genes we identified, WNT10B via the canonical WNT pathway and FGF2 via FGFR1 appear to regulate *Shh* induction and the subsequent patterning process.

WNT signaling may be the key for the dorsal properties in limb formation. It has been reported that canonical WNT/β-catenin signaling plays essential roles in limb regeneration among vertebrates, including axolotls (*Kawakami et al., 2006*; *Lovely et al., 2022*). In the present study, we demonstrated that introducing *Wnt10b* or BIO treatment could induce *Shh* expression in VentBL, facilitating limb patterning. In mouse limb development, *Wnt10b* is expressed in the AER (*Witte et al., 2009*), and mutations in *Wnt10b* are associated with Split-Hand/Foot Malformation (*Ugur and Tolun, 2008*; *Al Ghamdi et al., 2020*; *Bilal et al., 2023*). However, there is no evidence suggesting that *Wnt10b* contributes to dorsal specificity during limb development. This raises questions about whether the function of *Wnt10b* is unique to axolotls or whether its orthologs in other species might share functional redundancy with other WNT genes. Further studies are needed to clarify this point. Notably, WNT10B utilizes the canonical Wnt signaling pathway, similar to WNT7A, a well-known WNT family gene critical for dorsoventral limb patterning in amniotes. In amniotes, *Wnt7a* is expressed in the dorsal ectoderm of developing limb buds and induces *Lmx1b* expression in dorsal mesenchyme, thereby establishing dorsal characteristics (*Riddle et al., 1995*; *Cygan et al., 1997*; *Chen and Johnson, 2002*). In axolotls, dorsal-specific *Wnt7a* expression has not been confirmed. Our RNA-seq analysis showed that there was almost no significant difference in *Wnt7a* expression levels between DorBL and VentBL ($\log_2$ (normalized counts) = 5.97, $\log_2$ (fold change) = –0.810, p = 0.401), consistent with previous studies (*Shimokawa et al., 2013*). Similarly, there was no significant difference in *En1* expression ($\log_2$ (normalized counts) = 0.909, $\log_2$ (fold change) = –1.812, p = 0.264). In amniote limb development, *En1* is expressed in the ventral ectoderm, where it restricts *Wnt7a* expression to the dorsal ectoderm and thereby prevents induction of *Lmx1b* in the ventral mesenchyme (*Loomis et al., 1996*; *Logan et al., 1997*; *Chen and Johnson, 2002*). These results suggest that *Wnt7a* does not

have a dorsal-specific function, at least in axolotl limb regeneration. While WNT10B could function as a dorsal-mediated signal, WNT10B is unlikely to induce dorsal identity, as ectopic *Lmx1b* expression was not observed in *Wnt10b*-introduced VentBL, which formed double-ventral limbs (*Figure 5F–H*). This indicates that *Wnt10b* in axolotl limb regeneration does not simply replace the function of *Wnt7a* in amniote limb development. Nevertheless, the involvement of canonical WNT signaling in dorsal function and limb morphogenesis remains an important area for further investigation.

We identified FGF2 as the ventral-mediated signal, which plays a crucial role in the limb patterning process during axolotl limb regeneration. This aligns with previous studies showing that *Fgf2* expression correlates with limb regeneration in salamanders (*Giampaoli et al., 2003*). *Fgf2* expression was relatively high in VentBL, and *Fgf2*-introduced DorBL formed patterned limbs (*Figure 5*). These results indicate that *Fgf2* overexpression can substitute for the presence of ventral cells in the limb patterning process. It is noteworthy that FGF2 application does not appear to induce ventral identity, as the limbs formed from *Fgf2*-introduced DorBL exhibited dorsally symmetric limb structures. The use of FGF signaling as a ventral-mediated output downstream of dorsoventral identity has not been documented in other species examined to date. Whether this represents an axolotl-specific regulatory mechanism or a broader function of FGF signaling in limb morphogenesis remains to be determined.

Our findings suggest that although WNT10B and FGF2 act as dorsal- and ventral-mediated signals, they do not alter dorsal or ventral identity itself. In amniote limb development, WNT7A and EN1 regulate dorsoventral identity through *Lmx1b* expression. In contrast, in axolotls, *Wnt10b*-electroporation in VentBL or *Fgf2*-electroporation in DorBL did not affect the expression patterns of *Lmx1b* (*Figure 5C*). Moreover, the limbs formed from such DorBL and VentBL exhibited dorsally or ventrally symmetric structures (*Figure 5H–J*, *Figure 5—figure supplement 2*). These findings suggest that dorsoventral identities in axolotls are not affected by *Wnt10b* or *Fgf2* overexpression. We previously reported that the expression patterns of *Lmx1b* in axolotl limb regeneration are likely to depend on the positional origins of cells (*Iwata et al., 2020*; *Yamamoto et al., 2022*). The identities of cells along the dorsoventral axis may be controlled by their positional memory and determined before the initiation of limb regeneration.

The present study revealed that the dorsal- and ventral-mediated signals WNT10B and FGF2 regulate *Shh* expression during limb patterning in axolotls. These findings highlight a degree of conservation between axolotl limb regeneration and amniote limb development. In amniote limb development, FGF and WNT signaling pathways are key regulators of *Shh* expression. In developing amniote limb buds, *Fgf2* is expressed in the ectoderm, including the AER, and in adjacent mesoderm, and promotes distal outgrowth. FGFs, including FGF2, supplied from the AER are known to maintain *Shh* expression (*Laufer et al., 1994*; *Niswander et al., 1994*; *Yang and Niswander, 1995*; *Li et al., 1996*). Similarly, WNT7A regulates *Shh* expression in amniote limb development, and loss of WNT7A function results in reduced *Shh* expression in the zone of polarizing activity and the deletion of posterior structures (*Parr and McMahon, 1995*). Thus, our findings provide new insight into both conserved and divergent aspects of dorsal- and ventral-mediated signaling in the regulation of *Shh* expression, thereby furthering our understanding of limb morphogenesis.

In this study, RNA-seq analysis of ALM blastemas induced on the dorsal or ventral side listed *Wnt10b* and *Fgf2* as genes that are more highly expressed in DorBL and VentBL, respectively. In the ALM context, *Wnt10b* or *Fgf2* overexpression was sufficient to substitute for dorsal or ventral tissues, respectively, and to drive *Shh* induction and subsequent limb patterning even in the absence of those tissues (*Figure 5*). However, in amputation-induced blastemas during normal regeneration, ISH did not reveal clear expression patterns for *Wnt10b* or *Fgf2* (*Figure 5—figure supplement 3A*), and reanalysis of single-cell RNA-seq from the regenerating blastema (*Li et al., 2021*) showed that their expression did not strictly follow *Lmx1b* expression (*Figure 5—figure supplement 4J, K*). These results may partially reflect technical bias; low-level signals of epithelial and CT/fibroblast markers can be detected outside their expected clusters (*Figure 5—figure supplement 4A, B*). Moreover, because we focused our reanalysis on the 7 dpa (MB) sample—guided by our bulk RT-qPCR data suggesting that the expression of these gene peaks at the MB stage (*Figure 5—figure supplement 3C*)—it remains possible that clearer, and potentially different, expression patterns could be observed if other datasets are used. By contrast, and consistent with our bulk RNA-seq results, RT-qPCR of manually microdissected dorsal and ventral halves of regenerating blastemas showed that *Wnt10b* and *Fgf2* were expressed at significantly higher levels in the dorsal and ventral halves, respectively

(*Figure 5—figure supplement 3B*). These results suggest that *Wnt10b* expression and *Fgf2* expression are mediated by dorsal and ventral cells, respectively, but their expression is not restricted to dorsal or ventral cells. Our results on *Wnt10b*-electroporated VentBL and *Fgf2*-electroporated DorBL suggest that both activation of WNT10B and FGF2 is required for *Shh* induction and proceeds with limb patterning. To fully understand axolotl limb regeneration, it will be important to determine how *Wnt10b* and *Fgf2* expression is regulated, how their downstream programs are deployed, and how dorsal and ventral cells respond to these signals in future studies.

### The dorsoventral-mediated Shh induction mechanism

In the present study, we found the importance of the co-presence of cells carrying the anterior, posterior, dorsal, and ventral identity for successful axolotl limb patterning (*Figure 7*). In our proposed model, following amputation, nerves first trigger blastema induction by secreting nerve factors, such as BMPs and FGFs (*Makanae et al., 2014a*, *Satoh et al., 2016*). These nerve factors stimulate connective tissue cells, including dermal cells, to generate multipotent mesenchymal blastemal cells (*Muneoka et al., 1986*; *Kragl et al., 2009*; *Hirata et al., 2010*; *Gerber et al., 2018*). Within the induced blastema, *Wnt10b* and *Fgf2* expression are mediated by the dorsal and ventral cells, respectively. In the next phase, the co-existence of WNT10B and FGF2 signaling induces *Shh* expression in the posterior region of the blastema. During this phase, *Fgf8* is expressed in the anterior mesenchyme independently of the co-existence of dorsal and ventral cells (*Figure 7A*). These expression patterns of *Wnt10b*, *Fgf2*, *Shh*, and *Fgf8* may be mediated by the positional memory of cells (*Otsuki and Tanaka, 2022*). In the subsequent phase, the anteroposterior interaction, mediated by FGF8 and SHH, supports distal outgrowth to form a limb. This model explains previous observations in studies on double-half limbs and ALM blastemas (AntBL, PostBL, DorBL, and VentBL), which typically fail to form a limb. In blastemas induced from double-anterior and double-posterior limbs, or AntBL (*Figure 7B*) and PostBL (*Figure 7C*), SHH or FGF8 proteins are likely absent due to the lack of posteriorly or anteriorly derived cells, respectively. Similarly, in blastemas induced by amputating double-dorsal and double-ventral limbs, or DorBL (*Figure 7D*) and VentBL (*Figure 7E*), FGF2 or WNT10B proteins are likely absent or insufficient because of the lack of ventrally or dorsally derived cells, respectively. This results in the absence of SHH, even if posteriorly derived cells are present. In all these cases, the absence of either FGF8 or SHH disrupts the anteroposterior interaction, leading to failure in limb patterning. We conclude that the requirement for cells derived from all four positional origins is underpinned by this model. In this model, dorsal- and ventral-mediated signals activate the posterior SHH, enabling mutual interaction with the anterior FGF8. This interplay ensures proper anteroposterior interaction and complete limb patterning. Our findings contribute to understanding how the integration of four positional identities—dorsal, ventral, anterior, and posterior—drives proper limb patterning during axolotl limb regeneration.

## Materials and methods

**Key resources table**

| Reagent type (species) or resource | Designation | Source or reference | Identifiers | Additional information |
|---|---|---|---|---|
| Gene (*Ambystoma mexicanum*) | WNT4 | AXOLOTL-OMICS | AMEX60DD052091 | |
| Gene (*Ambystoma mexicanum*) | WNT10B | AXOLOTL-OMICS | AMEX60DD029981 | |
| Gene (*Ambystoma mexicanum*) | TGFB2 | AXOLOTL-OMICS | AMEX60DD036126 | |
| Gene (*Ambystoma mexicanum*) | FGF2 | AXOLOTL-OMICS | AMEX60DD044865 | |
| Gene (*Ambystoma mexicanum*) | FGF7 | AXOLOTL-OMICS | AMEX60DD003767 | |

*Continued on next page*

*Continued*

| Reagent type (species) or resource | Designation | Source or reference | Identifiers | Additional information |
|---|---|---|---|---|
| Genetic reagent (*Ambystoma mexicanum*) | Wild type (leucistic) | Hiroshima University Amphibian Research Center | Wild type (leucistic) | Procured from the Hiroshima University Amphibian Research Center |
| Antibody | anti-GFP (rabbit IgG, polyclonal) | MBL | RRID:AB_591819 | (1:500) |
| Antibody | anti-acetylated alpha tubulin (mouse IgG, monoclonal) | Santa Cruz | RRID:AB_628409 | (1:1000) |
| Antibody | anti-mouse IgG Alexa 488 (goat IgG, polyclonal) | Invitrogen | RRID:AB_143160 | (1:1000) |
| Antibody | anti-rabbit IgG Alexa 488 (donkey IgG, polyclonal) | Invitrogen | RRID:AB_2535792 | (1:500) |
| Recombinant DNA reagent | *Wnt10b*-p2a-AcGFP-pCS2 (plasmid) | This paper | N/A | Available from our group upon request |
| Recombinant DNA reagent | *Wnt4*-p2a-AcGFP-pCS2 (plasmid) | This paper | N/A | Available from our group upon request |
| Recombinant DNA reagent | *Fgf2*-p2a-AcGFP-pCS2 (plasmid) | This paper | N/A | Available from our group upon request |
| Recombinant DNA reagent | *Fgf7*-p2a-AcGFP-pCS2 (plasmid) | This paper | N/A | Available from our group upon request |
| Recombinant DNA reagent | *Tgfb2*-p2a-AcGFP-pCS2 (plasmid) | This paper | N/A | Available from our group upon request |
| Peptide, recombinant protein | Bmp2 | R&D Systems | Cat#355-BM | |
| Peptide, recombinant protein | Fgf2 | R&D Systems | Cat#3139-FB | |
| Peptide, recombinant protein | Fgf8 | R&D Systems | Cat#423-F8 | |
| Commercial assay or kit | MS222 | Sigma-Aldrich | Cat#A5042 | |
| Commercial assay or kit | FEWBlue TA PCR Cloning Kit, pTAC-2 | BioDynamics Lab Inc | Cat#DS126 | |
| Commercial assay or kit | SP6 RNA Polymerase | Takara | Cat#2520A | |
| Commercial assay or kit | T7 RNA Polymerase | Takara | Cat#2540A | |
| Commercial assay or kit | PrimeScript II 1st strand cDNA Synthesis Kit | Takara | Cat#6210A | |
| Commercial assay or kit | KAPA SYBR Fast qPCR Kit | Kapa Biosystems | Cat#KK4605 | |
| Commercial assay or kit | Genopure Maxi kit | Roche | Cat#03143422001 | |
| Commercial assay or kit | In-Fusion HD Cloning Kit | Clontech | Cat#639648 | |
| Chemical compound, drug | 6-Bromoindirubin-3-oxime (BIO) | Selleck | Cat#S7198 | |
| Software, algorithm | StepOne Software v2.1 system | Thermo Fisher Scientific | RRID:SCR_023455 | |
| Software, algorithm | ilastik | *Berg et al., 2019* | RRID:SCR_015246 | |
| Software, algorithm | Seurat | *Hao et al., 2021* | RRID:SCR_016341 | |
| Other | Proteinase K | Invitrogen | Cat#25530049 | |
| Other | 5-Bromo-4-chloro-3-indolyl Phosphate p-Toluidine Salt (BCIP) | Nacalai | Cat#05643-11 | |
| Other | Nitro Blue Tetrazolium (NBT) | Nacalai | Cat#24720-01 | |
| Other | Hoechst | Nacalai | Cat#19172-51 | (5 µl/40 ml TBST) |
| Other | TriPure reagent | Roche | Cat#11667157001 | |

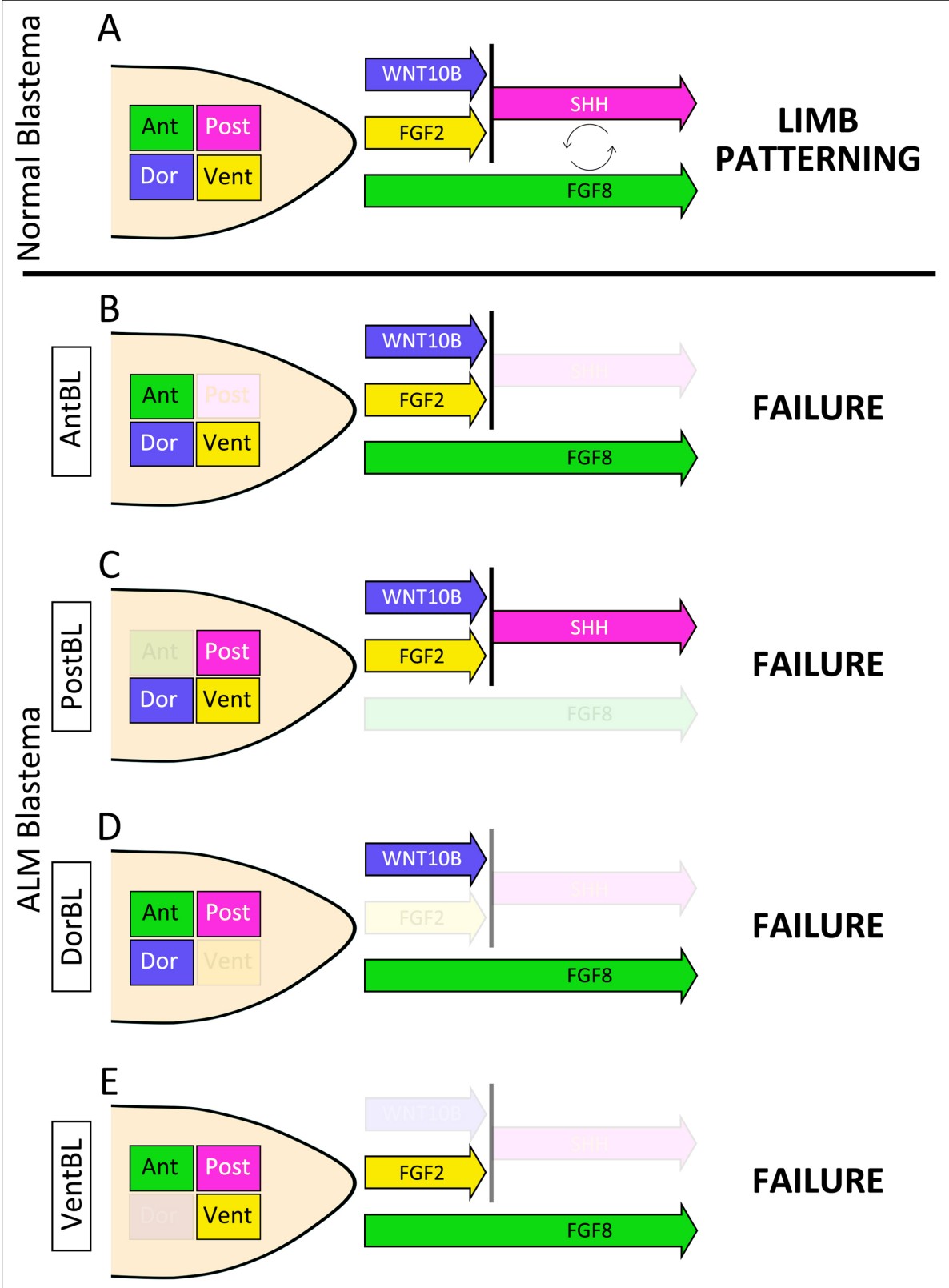

**Figure 7.** The dorsoventral-mediated Shh induction mechanism. Schematic images of a normal blastema (**A**), AntBL (**B**), PostBL (**C**), DorBL (**D**), and VentBL (**E**). Green, red, blue, and yellow boxes within the blastema represent cells derived from anterior, posterior, dorsal, and ventral regions, respectively. Colored arrows indicate the presence of the corresponding signals mediated by these cells. In this model, limb patterning requires both FGF8 and SHH, and *Shh* expression in posteriorly derived cells is induced by the co-existence of WNT10B and FGF2.

## Animal procedures

Axolotls between 4 and 15 cm from snout to tail tip were housed in tap water at 22°C. Both forelimbs and hind limbs were used for surgical procedures without distinction. We did not specifically distinguish between the sexes of the animals. This is because there is no evidence of gender-based differences in limb regeneration. Transgenic axolotls were obtained from the Ambystoma Genetic Stock Center (http://www.ambystoma.org/genetic-stock-center). Animals were anesthetized in 0.1% MS222 (Sigma-Aldrich) solution before surgical procedures.

In the ALM experiments, skin was peeled from the anterior, posterior, dorsal, or ventral side of a limb at the stylopod level, so that the skin of the opposite side was not injured. Thus, the size of the injured area depended on the limb size. Then, thick bundles of nerve trunks running the center of a limb were deviated to the injured region. For the cell-tracing experiment, a piece of posterior half of the dorsal or ventral skin was obtained from GFP-expressing transgenic animals and grafted to the ALM region.

Protein-soaked beads for grafting were prepared as previously described (*Makanae et al., 2014b*; *Kashimoto et al., 2023*). In brief, air-dried gelatin beads were allowed to swell in stock solution (1 μg/μl) prepared following the manufacturer's instructions. Equal amounts of proteins were used when formulating the combination protein mixture. Beads were soaked in the mixture of proteins (Bmp2 [mouse], Fgf2 [mouse], and Fgf8 [human/mouse]; R&D Systems) overnight at 4°C. For control experiments, gelatin beads were soaked in DDW. Before grafting, dorsal or ventral skin was peeled because full-thickness skin inhibits limb regeneration (*Thornton, 1962*; *Mescher, 1976*; *Tsai, 2020*). The beads were grafted under the wounded epidermis at 3 dps. The limbs were fixed 10 days after grafting. The details of grafting procedures are as previously described (*Makanae et al., 2014a*; *Kashimoto et al., 2023*).

To analyze skeletal patterns, induced limbs were stained with alcian blue and alizarin red. Samples were fixed in 100% ethanol for 1 day at room temperature, then stained with Alcian blue solution (Wako, pH 2.0) in 20% acetic acid (Nacalai Tesque) with 80% ethanol solution at 37°C overnight. Then, samples were washed in tap water several times and fixed in 10% Formaldehyde Neutral Buffer Solution (Nacalai Tesque) for 1 day. Samples were then stained with Alizarin red S (Nacalai Tesque) in the solution (4% KOH:10% Formaldehyde Neutral Buffer Solution = 2:3) at room temperature for 1 day. Finally, samples were placed in graded glycerol with 4% KOH for clearing.

For BIO treatment, 10 mM stock solution of 6-bromoindirubin-3-oxime (BIO, Selleck, S7198) dissolved in DMSO was stored in the dark at 4°C. Axolotls soon after surgery were raised in tap water with BIO solution (experimental) or with the same amount of solvent, DMSO (control), until 14 dps for RT-qPCR and 30 dps for phenotype analysis. The containers, including water and axolotls, were kept in the dark during BIO or DMSO treatment. For BIO treatment, axolotls between 4 and 5 cm from snout to tail tip were used.

The care and treatment of the animals in this study was carried out under protocols approved by the Animal Care and Use Committee of Okayama University (reference no. 2024252). Every possible measure was taken to minimize animal suffering, in line with the NIH Guide for the Care and Use of Laboratory Animals.

## Sectioning and histological staining

Samples were fixed in 4% PFA/PBS overnight at room temperature before sectioning. The fixed samples were embedded in O.C.T. compound (Sakura) following 30% sucrose/PBS treatment for approximately 12 hr at 4°C. Frozen sections of 14 μm thickness were prepared using Leica CM1850 (Nussloch). The sections were well dried under an air dryer and kept at –80°C until use.

Standard hematoxylin and eosin (HE) staining and Masson's trichrome staining were used for histological analysis. To visualize cartilage formation, Alcian blue staining was performed before HE staining. In brief, sections were washed in tap water several times to remove the O.C.T. compound. Then, the sections were stained with Alcian blue (Wako, pH 2.0), and then HE staining was performed. Trichrome stain (Masson) kit (Sigma, HT15-1KT) was used for trichrome staining. The stained sections were mounted using Softmount (Wako).

## In situ hybridization

ISH was performed as described previously (*Yamamoto et al., 2022*). For probe synthesis, target genes cloned on pTAC-2 plasmid (BioDynamics Lab Inc) were amplified by PCR using M13 primers.

PCR fragments were purified and used as an RNA probe template. RNA probe synthesis was performed with Sp6 or T7 RNA polymerase (Takara) for 3 hr, and RNA was hydrolyzed, depending on the length of targets. The sections were washed in PBT to remove the O.C.T. compound, treated with proteinase K (10 µg/ml) (Invitrogen)/PBT for 20 min at room temperature, washed in PBT, treated with 4% PFA/PBS for 20 min at room temperature, washed in PBT, and then probes were hybridized at 62.5°C for approximately 18 hr. The sections were washed in wash buffer 1 (formamide:$H_2O$:20× SSC [3 M NaCl:0.3 M sodium citrate, pH 5.0] = 2:1:1), and then in wash buffer 2 (formamide:$H_2O$:20× SSC = 5:1:4). The samples were then incubated with anti-digoxigenin-AP Fab fragments (Sigma-Aldrich, 1/1000) for 2 hr at room temperature. Samples were stained with BCIP (Nacalai Tesque) and NBT (Nacalai Tesque) in alkaline phosphatase buffer (0.1 M NaCl, 0.1 M Tris-HCl [pH 9.5], 0.1% Tween20) for 18 hr at room temperature after washing in TBST.

## Immunofluorescence

Immunofluorescence on sections was carried out based on a previous report (*Yamamoto et al., 2022*). The antibodies were as follows: anti-GFP (MBL, #598, 1/500), anti-acetylated alpha tubulin (Santa Cruz, #sc-23950, 1/1000), anti-mouse IgG Alexa 488 (Invitrogen, #A11017, 1/1000), and anti-rabbit IgG Alexa 488 (Invitrogen, #A21206, 1/500). Nuclei were stained with Hoechst (Nacalai Tesque), and images were captured using an Olympus BX51 system.

## RNA-seq analysis

Total RNA was extracted from DorBL and VentBL at 10 dps using TriPure reagent (Roche), following the manufacturer's instructions. Three biological replicates were prepared for both samples. 150 bp paired-end RNA-seq reads were obtained under contract to Rhelixa (Tokyo, Japan), using Illumina Nova Seq 6000 and SMART-Seq HT Plus Kit (#R400449). RNA-seq data were analyzed on Galaxy (https://usegalaxy.eu/root) as follows. Sequence reads were trimmed and the quality was filtered by Trimmomatic v0.39 (*Bolger et al., 2014*) with the following parameters (LEADING:20, TRAILING:20, SLIDINGWINDOW:4:15, and MINLEN:36). Axolotl genome data (AmexG_v6.0-DD) and annotation data (AmexT_v47-AmexG_v6.0-DD.gtf) were obtained from AXOLOTL-OMICS (https://www.axolotl-omics.org/) (*Schloissnig et al., 2021*). Mapping to the Axolotl genome (AmexG_v6.0-DD) was performed with HISAT2 v2.2.1 (*Kim et al., 2015*). Count data were obtained with featureCounts v2.0.8 (*Liao et al., 2014*) on the basis of the Axolotl gene model (AmexT_v47-AmexG_v6.0-DD.gtf). DEGs were identified with DESeq2 v2.11.40.8 (*Love et al., 2014*) (p < 0.05). Among identified DEGs (DorBL > VentBL; 762 genes, VentBL > DorBL; 513 genes), genes annotated as 'intercellular signaling molecule' were explored on PANTHER (https://www.pantherdb.org/; *Thomas et al., 2003*), and 21 genes (DorBL > VentBL; 5 genes, VentBL > DorBL; 16 genes) were identified as candidate genes of dorsal- and ventral-mediated signals, as shown below:

DorBL > VentBL:

WNT4 (AMEX60DD052091), WNT10B (AMEX60DD029981), SEMA3A (AMEX60DD023165), LECT2 (AMEX60DD041044), ANGPTL5 (AMEX60DD049512).

VentBL > DorBL:

FCN1 (AMEX60DD050926), CXCL14 (AMEX60DD028973), DNER (AMEX60DD002010), FGF7 (AMEX60DD003767), CXCL12 (AMEX60DD052412), CXCL8 (AMEX60DD044133), SEMA6A (AMEX60DD042813), ANGPTL2 (AMEX60DD050490), ANGPTL6 (AMEX60DD031854), FGL2 (AMEX60DD006387), VWDE (AMEX60DD022491), TGFB2 (AMEX60DD036126), FGF2 (AMEX60DD044865), EFNA4 (AMEX60DD014882), EFNA3 (AMEX60DD014867), ANGPTL1 (AMEX60DD018261).

## Cloning candidate genes and preparation of construct vectors for overexpression

Among the identified candidate genes, we focused on *Wnt10b*, *Wnt4*, *Fgf2*, *Fgf7*, and *Tgfb2*. We cloned these genes in pTAC-2 plasmids from cDNA obtained from blastemas. The following primers were used:

Wnt10b Fwd: ATGGCCCACAGCTCACCCTCCGACACC
Wnt10b Rev: TCACTTGCACACATTCACCCATTCTGTG
Wnt4 Fwd: ATGGATGCTCACGAAAGCAGCGTATATC

Wnt4 Rev: TCACCGGCAGGTGTGCATTTCTACCAC
Fgf2 Fwd: ATGGCGGCGGGGAGCATCACCACCTTGC
Fgf2 Rev: TCAACTCTTGGCCGACATGGGAAGGAAAAG
Fgf7 Fwd: ATGCGCAGATGGGTGCTAGCTTGGATC
Fgf7 Rev: TCATGTGTTATTGGATATACGCATTGGA
Tgfb2 Fwd: ATGAGATTACAATTACTGAGAAAAAAAAATG
Tgfb2 Rev: TTAGCTGCACTTGCAAGATTTTACAATCA

We then inserted these genes to p2a-AcGFP-pCS2 vectors with In-Fusion HD Cloning Kit (Clontech). The following primers were used for PCR before In-Fusion reaction:

pCS2 inverse Fwd: GCTACTAACTTCAGCCTGCTGAAGCAGG
pCS2 inverse Rev: CATCGATGGGATCCTGCAAAAAGAACAAGTAGCTT
Wnt10b Fwd: AGGATCCCATCGATGGCCCACAGCTCACCCTCCGA
Wnt10b Rev: GCTGAAGTTAGTAGCCTTGCACACATTCACCCA
Wnt4 Fwd: AGGATCCCATCGATGGATGCTCACGAAAGCAGCG
Wnt4 Rev: GCTGAAGTTAGTAGCCCGGCAGGTGTGCATTTCTA
Fgf2 Fwd: AGGATCCCATCGATGGCGGCGGGGAGCATCACCA
Fgf2 Rev: GCTGAAGTTAGTAGCACTCTTGGCCGACATGGGAA
Fgf7 Fwd: AGGATCCCATCGATGCGCAGATGGGTGCTAGCTTG
Fgf7 Rev: GCTGAAGTTAGTAGCTGTGTTATTGGATATACGCA
Tgfb2 Fwd: AGGATCCCATCGATGAGATTACAATTACTGAGAAA
Tgfb2 Rev: GCTGAAGTTAGTAGCGCTGCACTTGCAAGATTTTA

Finally, the following DNA constructs were obtained: *Wnt10b*-p2a-GFP (pCS2), *Wnt4*-p2a-GFP (pCS2), *Fgf2*-p2a-GFP (pCS2), *Fgf7*-p2a-GFP (pCS2), and *Tgfb2*-p2a-GFP (pCS2).

## Electroporation

Each DNA construct was injected directly into the target region. Immediately after injection, electric pulses were applied (20 V, 50-ms pulse length, 950-ms interval, 10 times) with NEPA21 (Nepa gene). The injected DNA constructs were as follows: pCS2-AcGFP, pCS2-*Shh*-p2a-AcGFP, *Wnt10b*-p2a-GFP (pCS2), *Wnt4*-p2a-GFP (pCS2), *Fgf2*-p2a-GFP (pCS2), *Fgf7*-p2a-GFP (pCS2), and *Tgfb2*-p2a-GFP (pCS2). All plasmids were purified using a Genopure Maxi kit (Roche). Electroporation was performed at –3, 4, and 7 dps, and samples were fixed at 10 dps for ISH. To analyze the phenotype of 90 dps samples, electroporation was performed at –3, 4, 7, 10, 15, 20, 25, and 30 dps.

## qRT-PCR

The procedures of qRT-PCR were previously described (*Yamamoto et al., 2022*). In brief, RT was performed using Prime Script II (Takara), and RT-qPCR was performed using KAPA SYBR FAST qPCR Master Mix (Kapa Biosystems) and StepOne (Thermo Fisher Scientific). For all biological replicates, at least four technical replicates were performed. Based on the QC criteria of the StepOne Software v2.1 system, measurements flagged as an outlier within a replicate group or showing multiple Tm peaks were excluded. Primers were as follows:

Ef-1α Fwd: AACATCGTGGTCATCGGCCAT
Ef-1α Rev: GGAGGTGCCAGTGATCATGTT
Shh Fwd: GCTCTGTGAAAGCAGAGAACTCG
Shh Rev: CGCTCCGTCTCTATCACGTAGAA
Axin2 Fwd: GGCACTGACTTATCCCCAGG
Axin2 Rev: GCATCATTGGCTGTCAACGG
Lef1 Fwd: CTACACCGAGATCAGCCACC
Lef1 Rev: GCTGTGGTAGGAGTTGTGGG
Lmx1b Fwd: CTGGTCCATGGCTACGATCT
Lmx1b Rev: TTAGCAGCAGAAACGGGACT
Wnt10b Fwd: CAGAAGAGACCCAGGTGCAG
Wnt10b Rev: CGAAGGCCCAAGATGTCTGT
Fgf2 Fwd: TCTTCCTTCGCATCAACCCC

Fgf2 Rev: TTTCATTGCCATCAACCGCC

RNA for *Figure 5—figure supplement 1E* was prepared from 1 μM BIO- or DMSO-treated VentBL at 14 dps. *Ef-1α* was used as the internal control.

## Pixel classification and calculating symmetry scores

A machine learning-based method was applied for pixel classification using ilastik software (downloaded on https://www.ilastik.org/). The details and workflows were previously described (*Berg et al., 2019*). Each pixel in the images was classified into five classes. The regions of the background (Class 1), cartilage (Class 2), muscle (Class 3), other connective tissue (Class 4), and epidermis (Class 5) were annotated for training data. Using these annotated regions as references, pixels were automatically classified into the respective classes. In this process, the same training data were used for images obtained from the same limb. Then, symmetry scores were calculated for each class individually and for the combined set of all classes. In this process, the external shape of the section, which could bend randomly during sample fixation process, would affect the symmetry scores if the entire region were used. To focus on the symmetry of the anatomical patterns, images with 400 μm width were prepared (*Figures 4L and 5E, F*). The symmetry scores of pixel-classified images were calculated using Python. First, the center of the dorsal end and ventral end was set as the axis of symmetry. Then, one side of the image was flipped. Next, color masks were generated for both sides of the image. These masks identified pixels that matched the specified color. Pixels were considered to match if their RGB values were within the given tolerance range for all three channels. In pixel comparison, the masks for one side and the other side were compared to calculate the following pixels:

Matching pixels: The number of pixels that match in both sides for the specified color.
Total pixels: The total number of pixels matching the color in either side.

The symmetry scores were computed as (number of matching pixels)/(total pixels in both sides). The scores were obtained from 12 areas of each group. For statistical analysis, a two-tailed Welch's *t*-test was used. In this analysis, each group was compared to the intact limb group because the intact limb should be set as a typical asymmetrical structure.

## Reanalyzing single-cell data

We reanalyzed published single-cell RNA-seq data from a 7 dpa (MB) blastema (*Li et al., 2021*, under accession code PRJNA589484, *Figure 5—figure supplement 4*). We constructed a transcriptome index for axolotl from the available genome assembly and gene annotation (AmexG_v6.0-DD genome and AmexT_v47-AmexG_v6.0-DD annotation, *Schloissnig et al., 2021*). The transcript set was used to build a salmon index with default k-mer length (31), and without decoy sequences, and gene-level UMI counts were generated using the Alevin module of salmon, which performs lightweight (quasi-) mapping of reads directly to the transcriptome index (*Patro et al., 2017*; *Srivastava et al., 2019*). The mapping was done on Galaxy (https://usegalaxy.eu). This procedure yielded a whitelist of 8925 barcodes, corresponding to putative cells, and produced a per-cell by per-gene UMI count matrix. According to the Alevin summary output, 77.4% of reads aligned to the indexed axolotl transcriptome after barcode correction and UMI deduplication. The resulting count matrix was exported in Matrix Exchange (MEX) format (matrix.mtx, barcodes.tsv, and features.tsv).

The data were then imported into R (RStudio environment) as a Seurat object, a data structure for scRNA-seq data (*Hao et al., 2021*). Cells were filtered based on standard quality-control metrics, excluding droplets with extremely low gene complexity, extremely high total UMI counts, or unusually high mitochondrial or ribosomal RNA content. For normalization and integration of cells into a shared space, counts were log-normalized using Seurat's NormalizeData (default LogNormalize method with a scale factor of 10,000), and highly variable genes (HVGs) were identified using FindVariableFeatures (vst method; 2000 features). The data were scaled with ScaleData, and principal component analysis (PCA) was performed on the HVGs (RunPCA, 50 components). The first 30 principal components were used to construct a nearest-neighbor graph (FindNeighbors) and to perform community detection-based clustering (FindClusters, resolution = 0.3). The same set of PCs (dims 1:30) was also used for nonlinear dimensionality reduction by UMAP (RunUMAP). Gene expression patterns were visualized using FeaturePlot with order = TRUE so that high-expressing cells are drawn on top of low- or non-expressing cells. We also generated two-gene 'co-expression' maps by classifying cells as expressing gene A only, gene B only, both, or neither, and overlaying these classes on the UMAP using DimPlot.

## Statistics and reproducibility

We did not employ strict statistical methods to determine the sample size, but given the high reproducibility of the results, we considered these sample sizes to be sufficient. The investigators were not blinded to allocation during the experiments or outcome assessment due to the nature of the sample preparation. In all experiments, strict randomization was not performed; however, the animals used were selected at random.

## Acknowledgements

We are grateful to R Iwata and T Satoh for supporting office work and animal housing. Animals were obtained through Hiroshima University Amphibian Research Center. This work is supported by the Japan Society for the Promotion of Science KAKENHI grant-in-aid for scientific research (B) (20H03264 and 24K02034 to AS) and by a Grant-in-Aid for Japan Society for the Promotion of 805 Science fellows (24KJ1718 to SY).

## Additional information

### Funding

| Funder | Grant reference number | Author |
|---|---|---|
| Japan Society for the Promotion of Science | 20H03264 | Akira Satoh |
| Japan Society for the Promotion of Science | 24K02034 | Akira Satoh |
| Japan Society for the Promotion of Science | 24KJ1718 | Sakiya Yamamoto |

The funders had no role in study design, data collection, and interpretation, or the decision to submit the work for publication.

### Author contributions

Sakiya Yamamoto, Conceptualization, Resources, Data curation, Formal analysis, Funding acquisition, Validation, Investigation, Visualization, Methodology, Writing – original draft, Writing – review and editing; Saya Furukawa, Ayaka Ohashi, Resources, Writing – review and editing; Mayuko Hamada, Methodology, Writing – review and editing; Akira Satoh, Conceptualization, Resources, Supervision, Funding acquisition, Visualization, Methodology, Project administration, Writing – review and editing

### Author ORCIDs

Sakiya Yamamoto https://orcid.org/0009-0004-6539-989X
Ayaka Ohashi https://orcid.org/0009-0002-2401-1735
Mayuko Hamada https://orcid.org/0000-0001-7306-2032
Akira Satoh https://orcid.org/0000-0001-9821-4290

### Ethics

The care and treatment of the animals in this study was carried out under protocols approved by the Animal Care and Use Committee of Okayama University (reference no. 2024252). Every possible measure was taken to minimize animal suffering, in line with the NIH Guide for the Care and Use of Laboratory Animals.

Reviewer #1 (Public review): https://doi.org/10.7554/eLife.106917.3.sa1
Reviewer #2 (Public review): https://doi.org/10.7554/eLife.106917.3.sa2
Reviewer #3 (Public review): https://doi.org/10.7554/eLife.106917.3.sa3
Author response https://doi.org/10.7554/eLife.106917.3.sa4

## Additional files

### Supplementary files
MDAR checklist

### Data availability
RNA-seq FASTQ files have been deposited in the DNA Data Bank of Japan (DDBJ; https://www.ddbj. nig.ac.jp/) under BioProject accession PRJDB38065 and DRA accession DRA023661.

The following dataset was generated:

| Author(s) | Year | Dataset title | Dataset URL | Database and Identifier |
|---|---|---|---|---|
| Yamamoto S, Satoh A | 2025 | RNA-seq of dorsally and ventrally blastemas induced by accessory limb model (ALM) at 10 days post surgery | https://ddbj.nig. ac.jp/search/ entry/bioproject/ PRJDB38065 | DNA Data Bank of Japan (DDBJ), PRJDB38065 |

The following previously published dataset was used:

| Author(s) | Year | Dataset title | Dataset URL | Database and Identifier |
|---|---|---|---|---|
| Li H, Wei X, Zhou L, Zhang W, Wang C, Guo Y, Li D, Chen J, Liu T, Zhang Y, Ma S, Wang C, Tan F, Xu J, Liu Y, Yuan Y, Chen L, Wang Q, Qu J, Shen Y, Liu S, Fan G, Liu L, Liu X, Hou Y, Liu GH, Gu Y, Xu X | 2021 | blastema 7dpa | https://www.ncbi. nlm.nih.gov/sra/ SRX7140465 | NCBI Sequence Read Archive, SRX7140465 |

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
