## [Editor Report · eLife Assessment]

This **fundamental** work by Yamamoto and colleagues advances our understanding of how positional information is coordinated between axes during limb outgrowth and patterning. They provide **convincing** evidence that the dorsal-ventral axis feeds into anterior-posterior signaling, and identify the responsible molecules by combining transplantations with molecular manipulations. This work will be of broad interest to regeneration, tissue engineering, and evolutionary biologists.

---

## [Referee Report · Reviewer #1 (Public review)]

Summary:

The manuscript by Yamamoto et al. presents a model by which the four main axes of the limb are required for limb regeneration to occur in the axolotl. A longstanding question in regeneration biology is how existing positional information is used to regenerate the correct missing elements. The limb provides an accessible experimental system by which to study the involvement of the anteroposterior, dorsoventral, and proximodistal axes in the regenerating limb. Extensive experimentation has been performed in this area using grafting experiments. Yamamoto et al. use the accessory limb model and some molecular tools to address this question. There are some interesting observations in the study. In particular, one strength the potent induction of accessory limbs in the dorsal axis with BMP2+Fgf2+Fgf8 is very interesting. Although interesting, the study makes bold claims about determining the molecular basis of DV positional cues, but the experimental evidence is not definitive and does not take into account the previous work on DV patterning in the amniote limb. Also, testing the hypothesis on blastemas after limb amputation would be needed to support the strong claims in the study.

Strengths:

The manuscript presents some novel new phenotypes generated in axolotl limbs due to Wnt signaling. This is generally the first example in which Wnt signaling has provided a gain of function in the axolotl limb model. They also present a potent way of inducing limb patterning in the dorsal axis by the addition of just beads loaded with Bmp2+Fgf8+Fgf2.

Comments on revised version:

Re-evaluation: The authors have significantly improved the manuscript and their conclusions reflect the current state of knowledge in DV patterning of tetrapod limbs. My only point of consideration is their claim of mesenchymal and epithelial expression of Wnt10b and the finding that Fgf2 and Wnt10b are lowly expressed. It is based upon the failed ISH, but this doesn't mean they aren't expressed. In interpreting the Li et al. scRNAseq dataset, conclusions depend heavily on how one analyzes and interprets it. The 7DPA sample shows a very low representation of epithelial cells compared to other time points, but this is likely a technical issue. Even the epithelial marker, Krt17, and the CT/fibroblast marker show some expression elsewhere. If other time points are included in the analysis, Wnt10b, would be interpreted as relatively highly expressed almost exclusively in the epithelium. By selecting the 7dpa timepoint, which may or may not represent the MB stage as it wasn't shown in the paper, the conclusions may be based upon incomplete data. I don't expect the authors to do more work, but it is worth mentioning this possibility. The authors have considered and made efforts to resolve previous concerns.

---

## [Referee Report · Reviewer #2 (Public review)]

Summary:

This study explores how signals from all sides of a developing limb, front/back and top/bottom, work together to guide the regrowth of a fully patterned limb in axolotls, a type of salamander known for its impressive ability to regenerate limbs. Using a model called the Accessory Limb Model (ALM), the researchers created early staged limb regenerates (called blastemas) with cells from different sides of the limb. They discovered that successful limb regrowth only happens when the blastema contains cells from both the top (dorsal) and bottom (ventral) of the limb. They also found that a key gene involved in front/back limb patterning, called Shh (Sonic hedgehog), is only turned on when cells from both the dorsal and ventral sides come into contact. The study identified two important molecules, Wnt10B and FGF2, that help activate Shh when dorsal and ventral cells interact. Finally, the authors propose a new model that explains how cells from all four sides of a limb, dorsal, ventral, anterior (front), and posterior (back), contribute at both the cellular and molecular level to rebuilding a properly structured limb during regeneration

Strengths:

The techniques used in this study, like delicate surgeries, tissue grafting, and implanting tiny beads soaked with growth factors, are extremely difficult, and only a few research groups in the world can do them successfully. These methods are essential for answering important questions about how animals like axolotls regenerate limbs with the correct structure and orientation. To understand how cells from different sides of the limb communicate during regeneration, the researchers used a technique called in situ hybridization, which lets them see where specific genes are active in the developing limb. They clearly showed that the gene Shh, which helps pattern the front and back of the limb, only turns on when cells from both the top (dorsal) and bottom (ventral) sides are present and interacting. The team also took a broad, unbiased approach to figure out which signaling molecules are unique to dorsal and ventral limb cells. They tested these molecules individually and discovered which could substitute for actual dorsal and ventral cells, providing the same necessary signals for proper limb development. Overall, this study makes a major contribution to our understanding of how complex signals guide limb regeneration, showing how different regions of the limb work together at both the cellular and molecular levels to rebuild a fully patterned structure.

Weaknesses:

Because the expressional analyses are performed on thin sections of regenerating tissue, in the original manuscript, they provided only a limited view of the gene expression patterns in their experiments, opening the possibility that they could be missing some expression in other regions of the blastema. Additionally, the quantification method of the expressional phenotypes in most of the experiments did not appear to be based on a rigorous methodology. The authors' inclusion of an alternate expression analysis, qRT-PCR, on the entire blastema helped validate that the authors are not missing something in the revised manuscript.

Overall, the number of replicates per sample group in the original manuscript was quite low (sometimes as low as 3), which was especially risky with challenging techniques like the ones the authors employ. The authors have improved the rigor of the experiment in the revised manuscript by increasing the number of replicates. The authors have not performed a power analysis to calculate the number of animals used in each experiment that is sufficient to identify possible statistical differences between groups. However, the authors have indicated that there was not sufficient preliminary data to appropriately make these quantifications.

Likewise, in the original manuscript, the authors used an AI-generated algorithm to quantify symmetry on the dorsal/ventral axis, and my concern was that this approach doesn't appear to account for possible biases due to tissue sectioning angles. They also seem to arbitrarily pick locations in each sample group to compare symmetry measurements. There are other methods, which include using specific muscle groups and nerve bundles as dorsal/ventral landmarks, that would more clearly show differences in symmetry. The authors have now sufficiently addressed this concern by including transverse sections of the limbs annd have explained the limitations of using a landmark-based approach in their quantification strategy.

---

## [Referee Report · Reviewer #3 (Public review)]

Summary:

After salamander limb amputation, the cross-section of the stump has two major axes: anterior-posterior and dorsal-ventral. Cells from all axial positions (anterior, posterior, dorsal, ventral) are necessary for regeneration, yet the molecular basis for this requirement has remained unknown. To address this gap, Yamamoto et al. took advantage of the ALM assay, in which defined positional identities can be combined on demand and their effects assessed through the outgrowth of an ectopic limb. They propose a compelling model in which dorsal and ventral cells communicate by secreting Wnt10b and Fgf2 ligands respectively, with this interaction inducing Shh expression in posterior cells. Shh was previously shown to induce limb outgrowth in collaboration with anterior Fgf8 (PMID: 27120163). Thus, this study completes a concept in which four secreted signals from four axial positions interact for limb patterning. Notably, this work firmly places dorsal-ventral interactions upstream of anterior-posterior, which is striking for a field that has been focussed on anterior-posterior communication. The ligands identified (Wnt10b, Fgf2) are different to those implicated in dorsal-ventral patterning in the non-regenerative mouse and chick models. The strength of this study is in the context of ALM/ectopic limb engineering. Although the authors attempt to assay the expression of Wnt10b and Fgf2 during limb regeneration after amputation, they were unable to pinpoint the precise expression domains of these genes beyond 'dorsal' and 'ventral' blastema. Given that experimental perturbations were not performed in regenerating limbs - almost exclusively under ALM conditions - this author finds the title "Dorsoventral-mediated Shh induction is required for axolotl limb regeneration" a little misleading.

Strengths:

(1) The ALM and use of GFP grafts for lineage tracing (Figures 1-3) take full advantage of the salamander model's unique ability to outgrow patterned limbs under defined conditions. As far as I am aware, the ALM has not been combined with precise grafts that assay 2 axial positions at once, as performed in Figure 3. The number of ALMs performed in this study deserves special mention, considering the challenging surgery involved.

(2) The authors identify that posterior Shh is not expressed unless both dorsal and ventral cells are present. This echoes previous work in mouse limb development models (AER/ectoderm-mesoderm interaction) but this link between axes was not known in salamanders. The authors elegantly reconstitute dorsal-ventral communication by grafting, finding that this is sufficient to trigger Shh expression (Figure 3 - although see also section on Weaknesses).

(3) Impressively, the authors discovered two molecules sufficient to substitute dorsal or ventral cells through electroporation into dorsal- or ventral- depleted ALMs (Figure 5). These molecules did not change the positional identity of target cells. The same group previously identified the ventral factor (Fgf2) to be a nerve-derived factor essential for regeneration. In Figure 6, the authors demonstrate that nerve-derived factors, including Fgf2, are alone sufficient to grow out ectopic limbs from a dorsal wound. Limb induction with a 3-factor cocktail without supplementing with other cells is conceptually important for regenerative engineering.

(4) The writing style and presentation of results is very clear.

Overall appraisal:

This is a logical and well-executed study that creatively uses the axolotl model to advance an important framework for understanding limb patterning. The relevance of the mechanisms to normal limb regeneration are not yet substantiated, in the opinion of this reviewer. Additionally, Wnt10b and Fgf2 should be considered molecules sufficient to substitute dorsal and ventral identity (solely in terms of inducing Shh expression). It is not yet clear whether these molecules are truly necessary (loss of function would address this).

Comments on revisions:

Congratulations - I still find this an elegant and easy-to-read study with significant implications for the field! Linking your mechanisms to normal limb regeneration (i.e. regenerating blastema, not ALM), as well as characterising the cell populations involved, will be interesting directions for the future.

---

## [Author Response]

The following is the authors’ response to the current reviews.

**Public Reviews:**

**Reviewer #1 (Public review):**
Summary:The manuscript by Yamamoto et al. presents a model by which the four main axes of the limb are required for limb regeneration to occur in the axolotl. A longstanding question in regeneration biology is how existing positional information is used to regenerate the correct missing elements. The limb provides an accessible experimental system by which to study the involvement of the anteroposterior, dorsoventral, and proximodistal axes in the regenerating limb. Extensive experimentation has been performed in this area using grafting experiments. Yamamoto et al. use the accessory limb model and some molecular tools to address this question. There are some interesting observations in the study. In particular, one strength the potent induction of accessory limbs in the dorsal axis with BMP2+Fgf2+Fgf8 is very interesting. Although interesting, the study makes bold claims about determining the molecular basis of DV positional cues, but the experimental evidence is not definitive and does not take into account the previous work on DV patterning in the amniote limb. Also, testing the hypothesis on blastemas after limb amputation would be needed to support the strong claims in the study.Strengths:The manuscript presents some novel new phenotypes generated in axolotl limbs due to Wnt signaling. This is generally the first example in which Wnt signaling has provided a gain of function in the axolotl limb model. They also present a potent way of inducing limb patterning in the dorsal axis by the addition of just beads loaded with Bmp2+Fgf8+Fgf2.Comments on revised version:Re-evaluation: The authors have significantly improved the manuscript and their conclusions reflect the current state of knowledge in DV patterning of tetrapod limbs. My only point of consideration is their claim of mesenchymal and epithelial expression of Wnt10b and the finding that Fgf2 and Wnt10b are lowly expressed. It is based upon the failed ISH, but this doesn't mean they aren't expressed. In interpreting the Li et al. scRNAseq dataset, conclusions depend heavily on how one analyzes and interprets it. The 7DPA sample shows a very low representation of epithelial cells compared to other time points, but this is likely a technical issue. Even the epithelial marker, Krt17, and the CT/fibroblast marker show some expression elsewhere. If other time points are included in the analysis, Wnt10b, would be interpreted as relatively highly expressed almost exclusively in the epithelium. By selecting the 7dpa timepoint, which may or may not represent the MB stage as it wasn't shown in the paper, the conclusions may be based upon incomplete data. I don't expect the authors to do more work, but it is worth mentioning this possibility. The authors have considered and made efforts to resolve previous concerns.

We are grateful for the constructive comments. As Reviewer #1 suggested, we noted that clearer expression patterns of *Wnt10b* and *Fgf2* may be detectable in scRNA-seq analyses at other stages, and we also clarified that low-level signals of epithelial and CT/fibroblast markers outside their expected clusters may reflect technical bias in the Discussion section. In addition, we agree with the reviewer’s point that our unsuccessful ISH experiments and the low abundance detected by RT-qPCR do not demonstrate absence of expression, and that conclusions from reanalyzing the Li et al. scRNA-seq dataset can depend strongly on analytical choices; therefore, while we focused on the 7 dpa sample because our RT-qPCR data suggested that *Wnt10b* and *Fgf2* may be most enriched around the MB stage (the original study refers to 7 dpa as MB), we explicitly acknowledged that analyzing a single time point—especially one with a low representation of epithelial cells—may yield incomplete or stage-biased interpretations, and that inclusion of additional datasets could reveal clearer and potentially different expression patterns in the Discussion section. We also tempered our wording regarding the inferred cellular sources to avoid over-interpretation based on the current data in the Results section.

**Reviewer #2 (Public review):**
Summary:This study explores how signals from all sides of a developing limb, front/back and top/bottom, work together to guide the regrowth of a fully patterned limb in axolotls, a type of salamander known for its impressive ability to regenerate limbs. Using a model called the Accessory Limb Model (ALM), the researchers created early staged limb regenerates (called blastemas) with cells from different sides of the limb. They discovered that successful limb regrowth only happens when the blastema contains cells from both the top (dorsal) and bottom (ventral) of the limb. They also found that a key gene involved in front/back limb patterning, called Shh (Sonic hedgehog), is only turned on when cells from both the dorsal and ventral sides come into contact. The study identified two important molecules, Wnt10B and FGF2, that help activate Shh when dorsal and ventral cells interact. Finally, the authors propose a new model that explains how cells from all four sides of a limb, dorsal, ventral, anterior (front), and posterior (back), contribute at both the cellular and molecular level to rebuilding a properly structured limb during regeneration.Strengths:The techniques used in this study, like delicate surgeries, tissue grafting, and implanting tiny beads soaked with growth factors, are extremely difficult, and only a few research groups in the world can do them successfully. These methods are essential for answering important questions about how animals like axolotls regenerate limbs with the correct structure and orientation. To understand how cells from different sides of the limb communicate during regeneration, the researchers used a technique called in situ hybridization, which lets them see where specific genes are active in the developing limb. They clearly showed that the gene Shh, which helps pattern the front and back of the limb, only turns on when cells from both the top (dorsal) and bottom (ventral) sides are present and interacting. The team also took a broad, unbiased approach to figure out which signaling molecules are unique to dorsal and ventral limb cells. They tested these molecules individually and discovered which could substitute for actual dorsal and ventral cells, providing the same necessary signals for proper limb development. Overall, this study makes a major contribution to our understanding of how complex signals guide limb regeneration, showing how different regions of the limb work together at both the cellular and molecular levels to rebuild a fully patterned structure.Weaknesses:Because the expressional analyses are performed on thin sections of regenerating tissue, in the original manuscript, they provided only a limited view of the gene expression patterns in their experiments, opening the possibility that they could be missing some expression in other regions of the blastema. Additionally, the quantification method of the expressional phenotypes in most of the experiments did not appear to be based on a rigorous methodology. The authors' inclusion of an alternate expression analysis, qRT-PCR, on the entire blastema helped validate that the authors are not missing something in the revised manuscript.Overall, the number of replicates per sample group in the original manuscript was quite low (sometimes as low as 3), which was especially risky with challenging techniques like the ones the authors employ. The authors have improved the rigor of the experiment in the revised manuscript by increasing the number of replicates. The authors have not performed a power analysis to calculate the number of animals used in each experiment that is sufficient to identify possible statistical differences between groups. However, the authors have indicated that there was not sufficient preliminary data to appropriately make these quantifications.Likewise, in the original manuscript, the authors used an AI-generated algorithm to quantify symmetry on the dorsal/ventral axis, and my concern was that this approach doesn't appear to account for possible biases due to tissue sectioning angles. They also seem to arbitrarily pick locations in each sample group to compare symmetry measurements. There are other methods, which include using specific muscle groups and nerve bundles as dorsal/ventral landmarks, that would more clearly show differences in symmetry. The authors have now sufficiently addressed this concern by including transverse sections of the limbs annd have explained the limitations of using a landmark-based approach in their quantification strategy.

We are grateful for the careful evaluation of the technical rigor and quantification. We have benefited from the reviewer’s earlier feedback, which guided revisions that improved the manuscript’s rigor and presentation.

**Reviewer #3 (Public review):**
Summary:After salamander limb amputation, the cross-section of the stump has two major axes: anterior-posterior and dorsal-ventral. Cells from all axial positions (anterior, posterior, dorsal, ventral) are necessary for regeneration, yet the molecular basis for this requirement has remained unknown. To address this gap, Yamamoto et al. took advantage of the ALM assay, in which defined positional identities can be combined on demand and their effects assessed through the outgrowth of an ectopic limb. They propose a compelling model in which dorsal and ventral cells communicate by secreting Wnt10b and Fgf2 ligands respectively, with this interaction inducing Shh expression in posterior cells. Shh was previously shown to induce limb outgrowth in collaboration with anterior Fgf8 (PMID: 27120163). Thus, this study completes a concept in which four secreted signals from four axial positions interact for limb patterning. Notably, this work firmly places dorsal-ventral interactions upstream of anterior-posterior, which is striking for a field that has been focussed on anterior-posterior communication. The ligands identified (Wnt10b, Fgf2) are different to those implicated in dorsal-ventral patterning in the non-regenerative mouse and chick models. The strength of this study is in the context of ALM/ectopic limb engineering. Although the authors attempt to assay the expression of Wnt10b and Fgf2 during limb regeneration after amputation, they were unable to pinpoint the precise expression domains of these genes beyond 'dorsal' and 'ventral' blastema. Given that experimental perturbations were not performed in regenerating limbs - almost exclusively under ALM conditions - this author finds the title "Dorsoventral-mediated Shh induction is required for axolotl limb regeneration" a little misleading.Strengths:(1) The ALM and use of GFP grafts for lineage tracing (Figures 1-3) take full advantage of the salamander model's unique ability to outgrow patterned limbs under defined conditions. As far as I am aware, the ALM has not been combined with precise grafts that assay 2 axial positions at once, as performed in Figure 3. The number of ALMs performed in this study deserves special mention, considering the challenging surgery involved.(2) The authors identify that posterior Shh is not expressed unless both dorsal and ventral cells are present. This echoes previous work in mouse limb development models (AER/ectoderm-mesoderm interaction) but this link between axes was not known in salamanders. The authors elegantly reconstitute dorsal-ventral communication by grafting, finding that this is sufficient to trigger Shh expression (Figure 3 - although see also section on Weaknesses).(3) Impressively, the authors discovered two molecules sufficient to substitute dorsal or ventral cells through electroporation into dorsal- or ventral- depleted ALMs (Figure 5). These molecules did not change the positional identity of target cells. The same group previously identified the ventral factor (Fgf2) to be a nerve-derived factor essential for regeneration. In Figure 6, the authors demonstrate that nerve-derived factors, including Fgf2, are alone sufficient to grow out ectopic limbs from a dorsal wound. Limb induction with a 3-factor cocktail without supplementing with other cells is conceptually important for regenerative engineering.(4) The writing style and presentation of results is very clear.Overall appraisal:This is a logical and well-executed study that creatively uses the axolotl model to advance an important framework for understanding limb patterning. The relevance of the mechanisms to normal limb regeneration are not yet substantiated, in the opinion of this reviewer. Additionally, Wnt10b and Fgf2 should be considered molecules sufficient to substitute dorsal and ventral identity (solely in terms of inducing Shh expression). It is not yet clear whether these molecules are truly necessary (loss of function would address this).Comments on revisions:Congratulations - I still find this an elegant and easy-to-read study with significant implications for the field! Linking your mechanisms to normal limb regeneration (i.e. regenerating blastema, not ALM), as well as characterising the cell populations involved, will be interesting directions for the future.

We are grateful for the constructive comments. To mitigate the concerns raised by Reviewer #3, we cited a previous study suggesting that ALM was used as the alternative experimental system for studying limb regeneration (Nacu et al., 2016, Nature, PMID: 27120163; Satoh et al., 2007, Developmental Biology, PMID: 17959163) in the Introduction section. We are confident that our ALM-based data provide a reasonable basis for understanding limb regeneration. We agree that there are important remaining questions—such as which cell populations express *Wnt10b* and *Fgf2* and how endogenous WNT10B and FGF2 signals induce *Shh* expression in normal regeneration—which should be investigated in future studies to deepen our understanding of limb regeneration.

The following is the authors’ response to the original reviews.

**Recommendations for the authors:**

**Reviewing Editor Comments:**
The authors should be commended for addressing this gap - how cues from the DV axis interact with the AP axis during limb regeneration. Overall, the concept presented in this manuscript is extremely interesting and could be of high value to the field. However, the manuscript in its current form is lacking a few important data and resolution to fully support their conclusions, and the following needs to be addressed before publication:(1) ISH data on Wnt10b and FGF2 from various regeneration time points are essential to derive the conclusion. Preferably multiplex ISH of Wnt10b/Fgf2/Shh or at least canonical ISH on serial sections to demonstrate their expression in dermis/epidermis and order of gene expression i.e. Shh is only expressed after expression of Wnt10b/FGF2. It would certainly help if this can also be shown in regular blastema.

We are grateful for the constructive suggestion on assessing Wnt10b and Fgf2 expression during regular regeneration, and we agree that clarifying their expression patterns in regular blastemas is important for strengthening the conclusions of our study. Because we cannot currently ensure sufficient sensitivity with multiplex FISH in our laboratory—partly due to high background—, we conducted conventional ISH on serial sections of regular blastemas at several time points (Fig. S5A). However, the expression patterns of Wnt10b and Fgf2 were not clear. To complement the ISH results, we performed RT-qPCR on microdissected dorsal and ventral halves of regular blastemas at the MB stage (Fig. S5B). We found that Wnt10b and Fgf2 were expressed at significantly higher levels in the dorsal and ventral halves, respectively, compared to the opposite half. This dorsal/ventral biased expression of Wnt10b/Fgf2 is consistent with our RNA-seq data. We further quantified expression levels of Wnt10b, Fgf2, and Shh across stages (intact, EB, MB, LB, and ED) and found that Wnt10b and Fgf2 peaked at the MB stage, whereas Shh peaked at the LB stage—consistent with the editor’s request regarding the order of gene expression (Fig. S5C). This temporal offset in upregulation supports our model. These results are now included in the revised manuscript (Line 294‒306).

To identify the cell types expressing Wnt10b or Fgf2, we analyzed published single-cell RNA-seq data (7 dpa blastema (MB), Li et al., 2021). As a result, Fgf2 expression was observed in the mesenchymal cluster, whereas Wnt10b expression was observed in both mesenchymal and epithelial clusters (Fig. S6). However, because only a small fraction of cells expressed Wnt10b, the principal cellular source of WNT10B protein remains unclear. The apparent low abundance likely contributes to the weak ISH signals and reflects current technical limitations. In addition, Wnt10b and Fgf2 expression did not follow Lmx1b expression (Fig. S6J, K), and Wnt10b and Fgf2 themselves were not exclusive (Fig. S6L). These results are now included in the revised manuscript (Line 307‒321). Together with the RT-qPCR data (Fig. S5B), these results suggest that Wnt10b and Fgf2 are not exclusively confined to purely dorsal or ventral cells at the single-cell level, even though they show dorsoventral bias when assessed in bulk tissue. These results suggest that Wnt10b/Fgf2 expression is not restricted to dorsal/ventral cells but mediated by dorsal/ventral cells, and co-existence of both signals should provide a permissive environment for Shh induction. Defining the precise spatial patterns of Wnt10b and Fgf2 in regular regeneration will therefore be an important goal for future work.

(2) Validation of the absence of gene expression via qRT PCR in the given sample will increase the rigor, as suggested by reviewers.

We thank for this important suggestion and agree that validation by qRT-PCR increases the rigor of our study. Accordingly, we performed RT-qPCR on AntBL, PostBL, DorBL, and VentBL to corroborate the ISH results. The results are now included in Fig. 2. We also verified by RT-qPCR that *Shh* expression following electroporation and the quantitative results are now provided in Fig. 5.

(3) Please increase n for experiments where necessary and mention n values in the figures.

We thank for this helpful comment and agree on the importance of providing sufficient sample sizes. Accordingly, we increased the n for the relevant experiments and have indicated the n values in the corresponding figure legends.

(4) Most comments by all three reviewers are constructive and largely focus on improving the tone and language of the manuscript, and I expect that the authors should take care of them.

We thank the reviewers for their constructive feedback on the tone and language of the manuscript. We have carefully revised the text according to each comment, and we hope these modifications have improved both clarity and readability.

In addition, in revising the manuscript we also refined the conceptual framework. Our new analysis of *Wnt10b* and *Fgf2* expression during normal regeneration suggests that these genes are not expressed in a strictly dorsal- or ventral-specific manner at the single-cell level. When these observations are considered together with (i) the RNA-seq comparison of dorsally and ventrally induced ALM blastemas, (ii) RT-qPCR of microdissected dorsal and ventral halves of regenerating blastemas, and (iii) the functional electroporation experiments, our interpretation is that *Wnt10b* and *Fgf2* act as dorsal- and ventral-mediated signals, respectively: their production is regulated by dorsal or ventral cells, and the presence of both signals is required to induce *Shh* expression. Given those, we now think our conclusion might be explained without using the confusing term, “positional cue”. Because the distinction between “positional cue” and “positional information” could be confusing as noted by the reviewers, we rewrote our manuscript without using “positional cue.*”*

**Reviewer #1 (Recommendations for the authors):**
(1) Line 61: More explanation for what a double-half limb means is needed.

We thank the reviewer for this suggestion. We have revised the manuscript (Line 73‒76). Specifically, we now explain that a double-dorsal limb, for example, is a chimeric limb generated by excising the ventral half and replacing it with a dorsal half from the contralateral limb while preserving the anteroposterior orientation.

(2) Line 63-65: "Such blastemas form hypomorphic, spike-like structures or fail to regenerate entirely." This statement does not represent the breadth of work on the APDV axis in limb regeneration. The cited Bryant 1976 reference tested only double-posterior and double-anterior newt limbs, demonstrating the importance of disposition along the AP axis, not DV. Others have shown that the regeneration of double-half limbs depends upon the age of the animal and the length of time between the grafting of double-half limbs and amputation. Also, some double-dorsal or double-ventral limbs will regenerate complete AP axes with symmetrical DV duplications (Burton, Holder, and Jesani, 1986). Also, sometimes half dorsal stylopods regenerate half dorsal and half ventral, or regenerate only half ventral, suggesting there are no inductive cues across the DV axis as there are along the AP axis. Considering this is the basis of the study under question, more is needed to convince that the DV axis is necessary for the generation of the AP axis.

We thank the reviewer for this detailed and constructive comment. We acknowledge that previous studies have reported a range of outcomes for double-half limbs. For example, Burton et al. (1986) described regeneration defects in double-dorsal (DD) and double-ventral (VV) limbs, although limb patterning did occur in some cases (Burton et al., 1986, Table 1). As the reviewer notes, regenerative outcomes depend on variables such as animal age and the interval between construction of the double-half limb and amputation, sometimes called the effect of healing time (Tank and Holder, 1978). Moreover, variability has been reported not only in DD/VV limbs but also in double-anterior (AA) and double-posterior (PP) limbs (e.g., Bryant, 1976; Bryant and Baca, 1978; Burton et al., 1986). In the revised manuscript, we have therefore modified the statement to avoid over-generalization and to emphasize that regeneration can be incomplete under these conditions (Line 76‒82). Importantly, in order to provide the additional evidence requested and to directly re-evaluate whether dorsal and ventral cells are required for limb patterning, we performed the ALM experiments shown in Fig. 1. The ALM system allows us to assess this question in a binary manner (regeneration vs. non-regeneration), thereby strengthening the rationale for our conclusions regarding the necessity of the APDV orientations. We also revised a sentence at the beginning of the Results section to emphasize this point (Line 139‒140).

(3) Line 71: These findings suggest that specific signals from all four positional domains must be integrated for successful limb patterning, such that the absence of any one of them leads to failure." I was under the impression that half posterior limbs can grow all elements, but half anterior can only grow anterior elements.

We thank the reviewer for this helpful clarification. As summarized by Stocum, half-limb experiments show that while some digit formation can occur, limb patterning remains incomplete in both anterior-half and posterior-half limbs in some cases (Stocum, 2017). We see this point as closely related to the broader question of whether proper limb patterning requires the integration of signals from all four positional domains. As noted in our response above, our ALM experiments in Fig. 1 were designed to test this point directly, and our data support the interpretation that cells from all four orientations are necessary for correct limb patterning.

(4) Line 79-81: This is stated later in lines 98-105. I suggest expanding here or removing it here.

We thank the reviewer for this suggestion. In the original version, lines 79–81 introduced our use of the terms “positional cue” and “positional information,” and this content partially overlapped with what later appeared in lines 98–105. In the revised manuscript, we have substantially rewritten this section (Line 82‒84), including the sentences corresponding to lines 79–81 in the original version, to remove the term “positional cue,” as explained in our response to the Editor’s comment (4); our revision reflects new analyses indicating that Wnt10b and Fgf2 appear not be strictly restricted to dorsal or ventral cell populations, and we now describe these factors as dorsal- or ventral-mediated signals that act across dorsoventral domains to induce Shh expression. Accordingly, we no longer maintain the original use of “positional cue” and “positional information.”

(5) Line 92 - 93: "Similarly, an ALM blastema can be induced in a position-specific manner along the limb axes. In this case, the induced ALM blastema will lack cells from the opposite side." This sentence is difficult to follow. Isn't it the same thing stated in lines 88-90?

We thank the reviewer for this comment. We revised the sentence to improve readability and to avoid redundancy with original Lines 88–90 (Line 104‒106).

(6) Line 107: I think the appropriate reference is McCusker et al., 2014 (Position-specific induction of ectopic limbs in non-regenerating blastemas on axolotl forelimbs), although Vieira et al., 2019 can be included here. In addition, Ludolph et al 1990 should be cited.

We thank the reviewer for this suggestion. We have added McCusker et al. (2014) and Ludolph et al. (1990) as references in the revised manuscript (Line 120‒121).

(7) Line 107-109: A missing point is how the ventral information is established in the amniote limb. From what I remember, it is the expression of Engrailed 1, which inhibits the ventral expression of Wnt7a, and hence Lmx1b. This would suggest that there is no secreted ventral cue. This is a relatively large omission in the manuscript.

We thank the reviewer for this comment. We agree that ventral fate in amniotes is specified by En1 in the ventral ectoderm, which represses Wnt7a and thereby prevents induction of Lmx1b; accordingly, a secreted ventral morphogen analogous to dorsal Wnt7a has not been established. We added this point to the revised Introduction (Line 61‒64).

By contrast, in axolotl limb regeneration, our previous work on Lmx1b expression suggests that DV identities reflect the original positional identity rather than being re-specified during regeneration (Yamamoto et al., 2022). Within this framework, our original use of the term “ventral positional cue” does not imply a ventral patterning morphogen in the amniote sense; rather, it denotes downstream signals induced by cells bearing ventral identity that are required for the blastema to form a patterned limb. This interpretation is consistent with classic studies on double-half chimeras and ectopic contacts between opposite regions (Iten & Bryant, 1975; Bryant & Iten, 1976; Maden, 1980; Stocum, 1982) as well as with our ALM data (Fig. 1). For this reason, we intentionally used the term “positional cues” to refer to signals provided by cells bearing ventral identity, which can be considered separable from the DV patterning mechanism itself, in the original text. As explained in our response to the Editor’s comment (4), we describe these signals as “signals mediated by dorsal/ventral cells,” rather than “positional cues” in the revised manuscript.

The necessity of dorsal- and ventral-mediated signals is supported by classic studies on the double-half experiment. In the non-regenerating cases, structural patterns along the anteroposterior axis appear to be lost even though both anterior and posterior cells should, in principle, be present in a blastema induced from a double-dorsal or double-ventral limbs. In limb development of amniotes, Wnt7a/Lmx1b or En-1 mutants show that limbs can exhibit anteroposterior patterning even when tissues are dorsalized or ventralized—that is, in the relative absence of ventral or dorsal cells, respectively (Riddle et al., 1995; Chen et al., 1998; Loomis et al., 1996). Taken together, axolotl limb regeneration, in which the presence of both dorsal and ventral cells plays a role in anteroposterior patterning, should differ from other model organisms. It is reasonable to predict the dorsal- and ventral-mediated signals in axolotl limb regeneration. We included this point in the revised manuscript (Line 82‒89). However, there is no evidence that these signals are secreted molecules. For this reason, we have carefully used the term “dorsal-/ventral-mediated signals” in the Introduction without implying secretion.

(8) Introduction - In general, the argument is a bit misleading. It is written as if it is known that a ventral cue is necessary, but the evidence from other animal models is lacking, from what I know. I may be wrong, but further argument would strengthen the reasoning for the study.

We thank the reviewer for this thoughtful comment. We agree that it should not read as if it is known that a ventral cue is necessary. In the revised Introduction, we have addressed this in several ways. First, as described in our response to comment (7), we now explicitly note that in amniote limb development ventral identity is specified by En1-mediated repression of Wnt7a, and that a secreted ventral morphogen equivalent to dorsal Wnt7a has not been established. Second, we removed the term “positional cue” and no longer present “ventral positional cue” as a defined entity. Instead, we use mechanistic phrasing such as “signals mediated by ventral cells” and “signals mediated by dorsal cells,” which does not assume that such signals are secreted morphogens or universally conserved. Third, we have reframed the role of dorsal- and ventral-mediated signals as a working hypothesis specific to axolotl limb regeneration, rather than as a general conclusion across model systems.

(9) Line 129: Remove "As mentioned before".

We thank the reviewer for this suggestion. We have removed the phrase “As mentioned before” in the revised manuscript (Line 143).

(10) Figure 1: Are Lmx1, Fgf8, and Shh mutually exclusive? Multiplexed FISH would provide this information, and is a relatively important question considering the strong claims in the study.

We thank the reviewer for raising this important point. As noted in our response to the editor’s comment, we cannot currently ensure sufficiently high detection sensitivity with multiplex FISH in our laboratory. However, based on previous reports (Nacu et al., 2016), Fgf8 and Shh should be mutually exclusive. In contrast, with respect to Lmx1b, our analysis suggests that its expression is not mutually exclusive with either Fgf8 or Shh, at least their expression domains. To confirm this, we analyzed the published scRNA-seq data and the results were added to the supplemental figure 6. Fgf8 and Shh were expressed in both Lmx1b-positive and Lmx1b-negative cells (Fig. S6H, I), but Fgf8 and Shh themselves were mutually exclusive (Fig. S6M). This point is now included in the revised manuscript (Line 314‒317).

(11) Results section and Figure 2: More evidence is needed for the lack of Shh expression ISH in tissue sections. Demonstrating the absence of something needs some qPCR or other validation to make such a claim.

We thank the reviewer for this suggestion. We performed qRT-PCR on ALM blastemas to complement the ISH data (Fig. 2).

(12) Line 179: I think they are likely leucistic d/d animals and not wild-type animals based upon the images.

We thank the reviewer for this observation. In the revised manuscript, we have corrected the description to “leucistic animals” (Line 194).

(13) Line 183-186: I'm a bit confused about this interpretation. If Shh turns on in just a posterior blastema, wouldn't it turn on in a grafted posterior tissue into a dorsal or ventral region? Isn't this independent of environment, meaning Shh turns on if the cells are posterior, regardless of environment?

Our interpretation is that only posterior-derived cells possess the competency to express Shh. In other words, whether a cell is capable of expressing Shh depends on its original positional identity (Iwata et al., 2020), but whether it actually expresses Shh depends on the environment in which the cell is placed. The results of Fig. 3E and G indicate that Shh activation is dependent on environment and that the posterior identity is not sufficient to activate Shh expression. We have revised the manuscript to emphasize this distinction more clearly (Line 198‒203).

(14) Figure 4: Do the limbs have an elbow, or is it just a hand?

We thank the reviewer for this thoughtful question. From the appearance, an elbow-like structure can occasionally be seen; however, we did not examine the skeletal pattern in detail because all regenerated limbs used for this analysis were sectioned for the purpose of symmetry evaluation, and we therefore cannot state this conclusively. While this is indeed an important point, analyzing proximodistal patterning would require a very large number of additional experiments, which falls outside the main focus of the present study. For this reason, and also to minimize animal use in accordance with ethical considerations, we did not pursue further experiments here. In response to this point, we have added a description of the skeletal morphology of ectopic limbs induced by BMP2+FGF2+FGF8 bead implantation (Fig. 6). In these experiments, multiple ectopic limbs were induced along the same host limb. In most cases, these ectopic limbs did not show fusion with the proximal host skeleton, similar to standard ALM-induced limbs, although in one case we observed fusion at the stylopod level. We now note this observation in the revised manuscript (Line 347‒354).

We regard the relationship between APDV positional information and proximodistal patterning as an important subject for future investigation.

(15) Line 203 - 237: I appreciate the symmetry score to estimate the DV axis. Are there landmarks that would better suggest a double-dorsal or double-ventral phenotype, like was done in the original double-half limb papers?

We thank the reviewer for this thoughtful comment. In most cases, the limbs induced by the ALM exhibit abnormal and highly variable morphologies compared to normal limbs, making it difficult to apply consistent morphological landmarks as used in the original double-half limb studies. For this reason, we focused our analysis on “morphological symmetry” as a quantitative measure of DV axis patterning, and we have added this explanation to the manuscript (Line 232‒235). Additionally, we provided transverse sections along the proximodistal axis as supplemental figures (Figs. S2 and S4). In addition to reporting the symmetry score, we have explicitly stated in the text that symmetry was also assessed by visual inspection of these sections.

(16) Line 245-247: The experiment was done using bulk sequencing, so both the epithelium and mesenchyme were included in the sample. The posterior (Shh) and anterior (Fgf8) patterning cues are mesenchymally expressed. In amniotes, the dorsal cue has been thought to be Wnt7a from the epithelium. Can ISH, FISH, or previous scRNAseq data be used to identify genes expressed in the mesenchyme versus epithelium? This is very important if the authors want to make the claim for defining "The molecular basis of the dorsal and ventral positional cues" as was stated by the authors.

We thank the reviewer for highlighting this important point. As the reviewer notes, our bulk RNA-seq data do not distinguish between epithelial and mesenchymal expression domains. As noted in our response to the editor’s comment, we performed ISH and qPCR on regular blastemas. However, these approaches did not provide definitive information regarding the specific cell types expressing Wnt10b and Fgf2. To complement this, we re-analyzed publicly available single-cell RNA-seq data (from Li et al., 2021). As a results, Fgf2 was expressed mainly by the mesenchymal cells, and Wnt10b expression was observed in both mesenchymal and epithelial cells. These results are now included in the revised manuscript (Line 294‒321) and in supplemental figures (Fig. S6, S7).

(17) Was engrailed 1, lmx1b, or Wnt7a differentially expressed along the DV axis, suggesting similar signaling between? Are these expressed in mesenchyme? Previous work suggests Wnt7a is expressed throughout the mesenchyme, but publicly available scRNAseq suggests that it is expressed in the epithelium.

We thank the reviewer for this important comment. As noted, the reported expression patterns of DV-related genes are not consistent across studies, which likely reflects the technical difficulty of detecting these genes with high sensitivity. In our own experiments, expression of DV markers other than Lmx1b has been very weak or unclear by ISH. Whether these genes are expressed in the epithelium or mesenchyme also appears to vary depending on the detection method used. In our RNA-seq dataset, Wnt7a expression was detected at very low levels and showed no significant difference along the DV axis, while En1 expression was nearly absent. We have clarified these results in the revised manuscript (Line 437‒441). Our reanalysis of the published scRNA-seq likewise detected Wnt7a in only a very small fraction of cells. Accordingly, we consider it premature to reach a definitive conclusion—such as whether Wnt7a is broadly mesenchymal or restricted to epithelium—as suggested in prior reports. We also note that whether Wnt7a is epithelial or mesenchymal does not affect the conclusions or arguments of the present study. Although the roles of Wnt7a and En1 in axolotl DV patterning are certainly important, we feel that drawing a definitive conclusion on this issue lies beyond the scope of the present study, and we have therefore limited our description to a straightforward presentation of the data.

(18) Line 247-249: The sentence suggests that all the ligands were tried. This should be included in the supplemental data.

We thank the reviewer for this clarification. In fact, we tested only Wnt4, Wnt10b, Fgf2, Fgf7, and Tgfb2, and all of these results are presented in the figures. To avoid misunderstanding, we have revised the text to explicitly state that our analysis focused on these five genes (Line 272‒274).

(19) Line 249: An n = 3 seems low and qPCR would be a more sensitive means of measuring gene induction compared to ISH. The ISH would confirm the qPCR results. Figure 5C is also not the most convincing image of Shh induction without support from a secondary method.

We have increased the sample size for these experiments (Line 277‒280). In addition, to complement the ISH results, we confirmed Shh induction by qPCR following electroporation of Wnt10b and Fgf2 (Fig. 5D, E). In addition, because Shh signal in the Wnt10b-electroporated VentBL images was particularly weak and difficult to discern, we replaced that panel with a representative example in which Shh signal is more clearly visible. These data are now included in the revised manuscript (Line 280‒282).

(20) Line 253: It is confusing why Wnt10b, but not Wnt4 would work? As far as I know, both are canonical Wnt ligands. Was Wnt7a identified as expressed in the RNAseq, but not dorsally localized? Would electroporation of Wnt7a do the same thing as Wnt10b and hence have the same dorsalizing patterning mechanisms as amniotes?

We thank the reviewer for raising this challenging but important question. Wnt10b was identified directly from our bulk RNA-seq analysis, as was Wnt4. The difference in the ability of Wnt10b and Wnt4 to induce Shh expression in VentBL may reflect differences in how these ligands activate downstream WNT signaling programs. WNT10B is a potent activator of the canonical WNT/β-catenin pathway (Bennett et al., 2005), although WNT10B has also been reported to trigger a β-catenin–independent pathway (Lin et al., 2021). By contrast, WNT4 can signal through both canonical and non-canonical (β-catenin–independent) pathways, and the balance between these outputs is known to depend on cellular context (Li et al., 2013; Li et al., 2019). Consistent with a requirement for canonical WNT signaling, we found that pharmacological activation of canonical WNT signaling with BIO (a GSK3 inhibitor) was also sufficient to induce Shh expression in VentBL. However, despite this, it is still unclear why Wnt10b, but not Wnt4, was able to induce Shh under our experimental conditions. One possible explanation is that different WNT ligands can engage the same receptors (e.g., Frizzled/LRP6) yet can drive distinct downstream transcriptional programs (This may depend on the state of the responding cells, as Voss et al. predicted), resulting in ligand-specific outputs (Voss et al., 2025). This point is now included in the revised discussion section (Line 402‒412). At present, we cannot distinguish between these possibilities experimentally, and we therefore refrain from making a stronger mechanistic claim.

With respect to Wnt7a, we detected Wnt7a expression at very low levels, and without a clear dorsoventral bias, in our RNA-seq analysis of ALM blastemas (we describe this point in Line 437‒440). This is consistent with previous work suggesting that axolotl Wnt7a is not restricted to the dorsal region in regeneration. Because of this low and unbiased expression, and because our data already implicated Wnt10b as a dorsal-mediated signal that can act across dorsoventral domains to permit Shh induction, we did not prioritize Wnt7a electroporation in the present study. We therefore cannot conclude whether Wnt7a would behave similarly to Wnt10b in this context.

Importantly, these uncertainties about ligand-specific mechanisms do not alter our main conclusion. Our data support the idea that a dorsal-mediated WNT signal (represented here by WNT10B and canonical WNT activation) and a ventral-mediated FGF signal (FGF2) must act together to permit Shh induction, and that the coexistence of these dorsal- and ventral-mediated signals is required for patterned limb formation in axolotl limb regeneration.

(21) Is canonical Wnt signaling induced after electroporation of Wnt10b or Wnt4? qPCR of Lef1 and axin is the most common way of showing this.

We thank the reviewer for this helpful suggestion. In addition to examining Shh expression, we also assessed canonical WNT signaling by qPCR analysis of Axin2 and Lef1 following Wnt10b electroporation. The data is now included in Fig. 5.

(22) Line 255-256: qPCR was presented for Figure 5D, but ISH was used for everything else. Is there a technical reason that just qPCR was used for the bead experiments?

We thank the reviewer for this helpful comment. In the original submission, our goal was to test whether treatment with commercial FGF2 protein or BIO could reproduce the results obtained by electroporation. In the revised manuscript, to avoid confusion between distinct experimental aims, we removed the FGF2–bead data from this section and instead used RT-qPCR to quantitatively corroborate Shh induction after electroporation (Fig. 5D–E). RT-qPCR provided a sensitive, whole-blastema readout and allowed a paired design (left limb: factor; right limb: GFP control) that increased statistical power while minimizing animal use. To address the reviewer’s point more directly, we additionally performed ISH for the BIO treatment and now include those results in Supplementary Figure 3 (Line 287‒288).

(23) Line 261-263: The authors did not show where Wnt10B or Fgf2 is expressed in the limb as claimed. The RNAseq was bulk, so ISH of these genes is needed to make this claim. Where are Wnt10b and Fgf2 expressed in the amputated limb? Do they show a dorsal (Wnt10b) and ventral (Fgf2) expression pattern?

We thank the reviewer for raising this important point. As noted in our response to the editor’s comment, we performed ISH on serial sections of regular blastemas at several time points (Fig. S5A). However, the expression patterns of Wnt10b and Fgf2 along the dorsoventral axis were not clear. To complement the ISH results, we performed RT-qPCR on microdissected dorsal and ventral halves of regular blastemas at the MB stage (Fig. S5B). We found that Wnt10b and Fgf2 were expressed at significantly higher levels in the dorsal and ventral halves, respectively, compared to the opposite half. This dorsal/ventral biased expression of Wnt10b/Fgf2 is consistent with our RNA-seq data. To identify the cell types expressing Wnt10b or Fgf2, we analyzed published single-cell RNA-seq data (7 dpa blastema (MB), Li et al., 2021). As a result, Fgf2 expression was observed in the mesenchymal cluster, whereas Wnt10b expression was observed in both mesenchymal and epithelial clusters (Fig. S6). However, because only a small fraction of cells expressed Wnt10b, the principal cellular source of WNT10B protein remains unclear. The apparent low abundance likely contributes to the weak ISH signals and reflects current technical limitations. In addition, Wnt10b and Fgf2 expression did not follow Lmx1b expression (Fig. S6J, K), and Wnt10b and Fgf2 themselves were not exclusive (Fig. S6L). Together with the RT-qPCR data (Fig. S5B), these results suggest that Wnt10b and Fgf2 are not exclusively confined to purely dorsal or ventral cells at the single-cell level, even though they show dorsoventral bias when assessed in bulk tissue, suggesting that Wnt10b/Fgf2 expression is not dorsal-/ventral-specific but mediated by dorsal/ventral cells. Defining the precise spatial patterns of Wnt10b and Fgf2 in regular regeneration will therefore be an important goal for future work. These points are now included in the revised manuscript (Line 485‒501).

(24) Line 266-288: The formation of multiple limbs is impressive. Do these new limbs correspond to the PD location they are generated?

We thank the reviewer for this interesting question. Interestingly, from our observations, there does appear to be a tendency for the induced limbs to vary in length depending on their PD location. The skeletal patterns of the induced multiple limbs are now included in Fig. 6. However, as noted earlier, the supernumerary limbs exhibit highly variable morphologies, and a rigorous analysis of PD correlation would require a large number of induced limbs. Since this lies outside the main focus of the present study, we have not pursued this point further in the manuscript.

(25) Line 288: The minimal requirement for claiming the molecular basis for DV signaling was identified is to ISH or multiplexed FISH for Wnt10b and Fgf2 in amputated limb blastemas to show they are expressed in the mesenchyme or epithelium and are dorsally and ventrally expressed, respectively. In addition, the current understanding of DV patterning through Wnt7a, Lmx1b, and En1 shown not to be important in this model.

We thank the reviewer for this comment and fully agree with the point raised. We would like to clarify that we are not claiming to have identified the molecular basis of DV patterning. As the reviewer notes, molecules such as Lmx1b, Wnt7a, and En1 are well identified in other animal models as key regulators of DV positional identity. There is no doubt that these molecules play central roles in DV patterning. However, in axolotl limb regeneration, clear DV-specific expression has not been demonstrated for these genes except for Lmx1b. Therefore, further studies will be required to elucidate the molecular basis of DV patterning in axolotls.

Our focus here is more limited: we aim to identify the molecular basis for the mechanisms in which positional domain-mediated signals (FGF8, SHH, WNT10B, and FGF2) regulate the limb patterning process, rather than the molecular basis of DV patterning. In fact, our results on Wnt10b and Fgf2 suggest that these genes did not affect dorsoventral identities.

We recognize that this distinction was not sufficiently clear in the original text, and we have revised the manuscript to describe DV patterning mechanisms in other animals and clarify that the dorsal- and ventral-mediated signals are distinct from DV patterning (Line 444‒450). At least, we avoid claiming that the molecular basis for DV signaling was identified.

(26) Line 335: References are needed for this statement. From what I found, Wnt4 can be canonical or non-canonical.

We thank the reviewer for this helpful comment. We have revised the manuscript (Line 404‒407). We added these citations at the relevant location and adjusted nearby wording to avoid implying pathway exclusivity, in alignment with our response to comment (20).

(27) Line 337-338: The authors cannot claim "that canonical, but not non-canonical, WNT signaling contributes to Shh induction" as this was not thoroughly tested is based upon the negative result that Wnt4 electroporation did not induce Shh expression.

We thank the reviewer for this important clarification. We agree that our data do not allow us to conclude that non-canonical WNT signaling in general does not contribute to Shh induction. Accordingly, we have removed the phrase “but not non-canonical” and revised the text to emphasize that, within the scope of our experiments, Shh induction was not observed following Wnt4 electroporation, whereas it was observed with Wnt10b.

(28) Line 345: In order to claim "WNT10B via the canonical WNT pathway...appears to regulate Shh expression" needs at least qPCR to show WNT10B induces canonical signaling.

We thank the reviewer for this comment. As noted in our response to comment (21), we also assessed canonical WNT signaling by qPCR analysis of Axin2 and Lef1 following Wnt10b electroporation (Line 282‒285).

(29) Lines 361-372: A few studies have been performed on DV patterning of the mouse digit regeneration in regards to Lmx1b and En1. It may be good to discuss how the current study aligns with these findings.

We appreciate the reviewer’s suggestion. As the reviewer refers, several studies have been performed on dorsoventral (DV) patterning in mouse digit tip regeneration in relation to Lmx1b and En1 (e.g., Johnson et al., 2022; Castilla-Ibeas et al., 2023). In the present study, however, our main conclusion is different in the scope of studies on mouse digit tip regeneration. We show that, in the axolotl, pre-existing dorsal and ventral identities (as reflected by dorsally derived and ventrally derived cells in the ALM blastema) are required together to induce Shh expression, and that this Shh induction in turn supports anteroposterior interaction at the limb level. This mechanism—dorsal-mediated and ventral-mediated signals acting in combination to permit Shh expression—does not have a clear direct counterpart in the mouse digit tip literature. Moreover, even with respect to Lmx1b, the two systems behave differently. In mouse digit tip regeneration, loss of Lmx1b during regeneration does not grossly affect DV morphology of the regenerate (Johnson et al., 2022). By contrast, in our axolotl ALM system, the presence or absence of Lmx1b-positive dorsal tissue correlates with the final dorsoventral organization of the induced limb-like structures (e.g., production of double-dorsal or double-ventral symmetric structures in the absence of appropriate dorsoventral contact). Thus, the role of dorsoventral identity in our model is directly tied to patterned limb outgrowth at the whole-limb scale, whereas in the mouse digit tip it has been reported primarily in the context of digit tip regrowth and bone regeneration competence, not robust DV repatterning (Johnson et al., 2022).

For these reasons, we believe that an extended discussion of mouse digit tip regeneration would risk implying a mechanistic equivalence between axolotl limb regeneration and mouse digit tip regeneration that is not supported by current data. Because the regenerative contexts differ, and because Lmx1b does not appear to re-establish DV patterning in the mouse regenerates (Johnson et al., 2022), we have chosen not to include an explicit discussion of mouse digit tip regeneration in the main text.

(30) Line 408-433: Although I appreciate generating a model, this section takes some liberties to tell a narrative that is not entirely supported by previous literature or this study. For example, lines 415-416 state "Wnt10b and Fgf2 are expressed at higher levels in dorsal and the ventral blastemal cells, respectively" which were not shown in the study or other studies.

We thank the reviewer for this important comment. We agree that the original model based on RNA-seq data overstated the evidence. To address this point experimentally, we examined Wnt10b and Fgf2 expression in regular blastemas (Supplemental Figure 5 and 6). Accordingly, our model is now framed as an inductive mechanism for Shh expression—supported by results in ALM (WNT10B in VentBL; FGF2 in DorBL) and by DV-biased expression. Concretely, the sentence previously paraphrased as “Wnt10b and Fgf2 are expressed at higher levels in dorsal and ventral blastemal cells, respectively” has been replaced with wording that (i) avoids single-cell DV specificity and (ii) emphasizes dorsal-/ventral-mediated regulation and the requirement for both signals to allow Shh induction (Line 510‒511).

**Reviewer #2 (Recommendations for the authors):**
(1) Introduction:The authors' definitions of positional cues vs positional information are a little hard to follow, and do not appear to be completely accurate. From my understanding of what the authors explain, "positional information" is defined as a signal that generates positional identities in the regenerating tissue. This is a somewhat different definition than what I previously understood, which is the intrinsic (likely epigenetic) cellular identity associated with specific positional coordinates. On the other hand, the authors define "positional cues" as signals that help organize the cells according to the different axes, but don't actually generate positional identities in the regenerating cells. The authors provide two examples: Wnt7a as an example of positional information, and FGF8 as a positional cue. I think that coording to the authors definitions, FGF8 (and probobly Shh) are bone fide positional cues, since both signals work together to organize the regenerating limb cells - yet do not generate positional identities, because ectopic limbs formed from blastemas where these pathways have been activated do not regenerate (Nacu et al 2016). However, I am not sure Wnt7a constitutes an example of a "positional information" signal, since as far as I know, it has not been shown to generate stable dorsal limb identities (that remain after the signal has stopped) - at least yet. If it has, the authors should cite the paper that showed this. I think that some sort of diagram to help define these visually will be really helpful, especially to people who do not study regenerative patterning.

We thank the reviewer for this thoughtful comment. We now agree with the reviewer that our use of “positional cue” and “positional information” may have been confusing. In the revision—and as noted in our response to the Editor’s comment (4)—we have removed the term “positional cue” and no longer attempt to contrast it with “positional information.” Instead, we adopt phrasing that reflects our data and hypothesis: during limb patterning, dorsal-mediated signals act on ventral cells and ventral-mediated signals act on dorsal cells to induce Shh expression. This wording avoids implying that these signals specify dorsoventral identity.

Regarding WNT7A, we agree it has not been shown to generate a stable dorsal identity after signal withdrawal. In the revised Introduction we therefore describe WNT7A in amniote limb development as an extracellular regulator that induces Lmx1b in dorsal mesenchyme (with En1 repressing Wnt7a ventrally), rather than labeling it as “positional information” in a strict, identity-imprinting sense. We highlight this contrast because, in our axolotl experiments, WNT10B and FGF2 did not alter Lmx1b expression or dorsal–ventral limb characteristics when overexpressed, consistent with the idea that they act downstream of DV identity to enable Shh induction, not to establish DV identity.

(2) Results:It would be helpful if the number of replicates per sample group were reported in the figure legends.

We thank the reviewer for this suggestion. In accordance with the comment, we have added the number of replicates (n) for each sample group in the figure legends.

Figure 2 shows ISH for A/P and D/V transcripts in different-positioned blastemas without tissue grafts. The images show interesting patterns, including the lack of Shh expression in all blastemas except in posterior-located blastemas, and localization of the dorsal transcript (Lmx1b) to the dorsal half of A or P located blastemas. My only concern about this data is that the expression patterns are described in only a small part of the ectopic blastema (how representative is it?) and the diagrams infer that these expression patterns are reflective of the entire blastema, which can't be determined by the limited field of view. It is okay if the expression patterns are not present in the entire blastema -in fact, that might be an important observation in terms of who is generating (and might be receiving) these signals.

We thank the reviewer for this insightful comment. Because *Fgf8* and *Shh* expression was detectable only in a limited subset of cells, the original submission included only high-magnification images. In response to the reviewer’s valid concern about representativeness, we have now added low-magnification overviews of the entire blastema as a supplemental figure (Fig. S1) and clarified in the figure legend that these expression patterns can be focal rather than pan-blastemal (Line 795‒796).

In Figure 3, they look at all of these expression patterns in the grafted blastemas, showing that Shh expression is only visible when both D and V cells are present in the blastema. My only concern about this data is that the number of replicates is very low (some groups having only an N=3), and it is unclear how many sections the authors visualized for each replicate. This is especially important for the sample groups where they report no Shh expression -I agree that it is not observable in the single example sections they provide, but it is uncertain what is happening in other regions of the blastema.

We thank the reviewer for this important comment. To increase the reliability of the results, we have increased the number of biological replicates in groups where n was previously low. For all samples, we collected serial sections spanning the entire blastema. For blastemas in which *Shh* expression was observed, we present representative sections showing the signal. For blastemas without detectable *Shh* expression, we selected a section from the central region that contains GFP-positive cells for the Figure. To make these points explicit, we have added the following clarification to the Fig. 3 legend (Line 811‒815).

Figure 4: Shh overexpression in A/P/D/V blastemas - expression induces ectopic limbs in A/D/V locations. They analyzed the symmetry of these regenerates (assuming that Do and V located blastemas will exhibit D/V symmetry because they only contain cells from one side of that axis. I am a little concerned about how the symmetry assay is performed, since oblique sections through the digits could look asymmetric, while they are actually symmetric. It is also unclear how the angle of the boxes that the symmetry scores were based on was decided - I imagine that the score would change depending on the angle. It also appears that the authors picked different digits to perform this analysis on the different sample groups. I also admit that the logic of classification scheme that the authors used AI to perform their symmetry scoring analysis (both in Figures 4 and 5) is elusive to me. I think it would have been more informative if the authors leveraged the structural landmarks, like the localization of specific muscle groups. (If this experiment were performed in WT animals, the authors could have used pigment cell localization)... or generate more proximal sections to look at landmarks in the zeugopod.

We thank the reviewer for these detailed comments regarding the symmetry analysis. Because reliance on a computed symmetry score alone could raise the concerns noted by the reviewer, we now provide transverse sections along the proximodistal axis as supplemental figures (Figs. S2 and S4). These include levels corresponding to the distal end of the zeugopod and the proximal end of the autopod. In addition to reporting the symmetry score, we have explicitly stated in the text that symmetry was also assessed by visual inspection of these sections.

As also noted in our response to Reviewer #1 (comment 15), ALM-induced limbs frequently exhibit abnormal and highly variable morphologies, which makes it difficult to use consistent anatomical landmarks such as particular digits or muscle groups. For this reason, we focused our analysis on morphological symmetry rather than landmark-based metrics, and we emphasize this rationale in the revised text (Line 232‒235).

Regarding the use of bounding boxes, this procedure was chosen to minimize the effects of curvature or fixation-induced distortion. For each section, the box angle was adjusted so that the outer contour (epidermal surface) was aligned symmetrically; this procedure was applied uniformly across all conditions to avoid bias. We analyzed multiple biological replicates in each group, which helps mitigate potential artifacts due to oblique sectioning. To further reduce bias, we increased the number of fields included in the analysis to n = 24 per group in the revised version.

In addition, staining intensity varied among samples, such that a region identified as “muscle” in one sample could be assigned differently in another if classification were based solely on color. To avoid this problem, we used a machine-learning classifier trained separately for each sample, allowing us to group the same tissues consistently within that sample irrespective of intensity differences. In the context of ALM-induced limbs, where stable anatomical landmarks are not available, we consider this strategy the most appropriate. We have added this rationale to the revised manuscript for clarity (Line 239‒247).

Figure 5: The number of replicates in sample groups is relatively low and is quite variable between groups (ranging between 3 and 7 replicates). Zoom in to visualize Shh expression is small relative to the blastema, and it is difficult to discern why the authors positioned the window where they did, and how they maintained consistency among their different sample groups. In the examples of positive Shh expression - the signal is low and hard to see. Validating these expression patterns using some sort of quantitative transcriptional assay (like qRTPCR) would increase the rigor of this experiment ... especially given that they will be able to analyze gene expression in the entire blastema as opposed to sections that might not capture localized expression.

We thank the reviewer for this important comment. To increase the rigor of these experiments, we have increased the number of biological replicates in groups where n was previously low. In addition, because *Shh* signal in the *Wnt10b*-electroporated VentBL images was particularly weak and difficult to discern, we replaced that panel with a representative example in which *Shh* signal is more clearly visible. We also validated the *Shh* expression for *Wnt10b*–electroporated VentBL and *Fgf2*–electroporated DorBL by RT-qPCR, which assesses gene expression across the entire blastema. These results are now included in Fig. 5 and Line 280‒282. Finally, we clarified in the figure legend how the “window” for imaging was chosen: for samples with detectable *Shh* expression, the window was placed in the region where the signal was observed; for conditions without detectable *Shh* expression, the window was positioned in a comparable region containing GFP-positive cells (Line 836‒839). These revisions are included in the revised manuscript.

Figure 6: They treat dorsal and ventral wounds with gelatin beads soaked in a combination of BMP2+FGF8 (nerve factors) and FGF2 proposed ventral factor). Remarkably, they observe ectopic limb expression in only dorsal wounds, further supporting the idea that FGF2 provides the "ventral" signal. They show examples of this impressive phenotype on limbs with multiple ectopic structures that formed along the Pr/Di axis. Including images of tubulin staining (as they have in Figures 1 and 2) to ensure that the blastemas (or final regenerates) are devoid of nerves. The authors' whole-mount skeletal staining which shows fusion of the ectopic humerus with the host humerus, is a phenotype associated with deep wounding, which could provide an opportunity for more cellular contribution from different limb axes.

We thank the reviewer for these constructive comments. As noted in the prior study, when beads are used to induce blastemas without surgical nerve orientation, fine nerve ingrowth can still occur (Makanae et al., 2014), and the induced blastemas are not completely devoid of nerves. While it is still uncertain whether these recruited nerves are functional after blastema induction, it is an important point, and we added sentences about this in the revised manuscript (Line 341‒345).

Regarding the skeletal phenotype, despite careful implantation to avoid injuring deep tissues, bead-induced ectopic limbs on the dorsal side occasionally displayed fusion of the stylopod with the host humerus—a phenotype associated with deep wounding, as the reviewer notes. This observation suggests that contributions from a broader cellular population cannot be excluded. However, because fusion was observed in only 1 of 16 induced limbs analyzed, and because ectopic limbs induced at the forearm (zeugopod) level did not exhibit such fusion (n=1/6 for stylopod-level inductions; n=0/10 for zeugopod-level inductions), we believe that our main conclusion remains valid. Because fusion is not a typical outcome, we now present representative non-fusion cases—including zeugopod-origin examples—in the figure (Fig. 6L1, L2), and we report the fusion incidence explicitly in the text (Line 350‒354). We also note in the revised manuscript that stylopod fusion can occur in a minority of cases (Line 347‒349).

Figure 7 nicely summarizes their findings and model for patterning.

We thank the reviewer for this positive comment.

The table is cut off in the PDF, so it cannot be evaluated at this time.

In our copy of the PDF, the table appears in full, so this may have been a formatting issue. We have carefully checked the file and ensured that the table is completely included in the revised submission.

There is a supplemental figure that doesn't seem to be referenced in the text.

The supplemental figure (Fig. S1 of the original manuscript) is referenced in the text, but it may have been overlooked. To improve clarity, we have expanded the description in the manuscript so that the supplemental figure is more clearly referenced (Line 285‒291).

(3) Materials and Methods:No power analysis was performed to calculate sample group sizes. The authors have used these experimental techniques in the past and could have easily used past data to inform these calculations.

We thank the reviewer for this important comment. We did not include a power analysis in the manuscript because this was the first time we compared *Shh* and other gene expression levels among ALM blastemas of different positional origins using RT-qPCR in our experimental system. As we did not have prior knowledge of the expected variability under these specific conditions, it was difficult to predetermine appropriate sample sizes.

**Reviewer #3 (Recommendations for the authors):**
General:Congratulations - I found this an elegant and easy-to-read study with significant implications for the field! If possible, I would urge you to consider adding some more characterisation of Wnt10b and Fgf2- which cell types are they expressed in? If you can link your mechanisms to normal limb regeneration too (i.e., regenerating blastema, not ALM), this would significantly elevate the interest in your study.

We sincerely thank the reviewer for these encouraging comments. As also noted in our response to the editor’s comment, we have analyzed the expression patterns of Wnt10b and Fgf2 in regular blastemas (Line 294‒306). Although clear specific expression patterns along dorsoventral axis were not detected by ISH, likely due to technical limitations of sensitivity, RT-qPCR revealed significantly higher expression levels of Wnt10b in the dorsal half and Fgf2 in the ventral half of a regular blastema (Fig. S5). In addition, we analyzed published single-cell RNA-seq data (7 dpa blastema, Li et al., 2021) (Line 307‒321). As a result, Fgf2 expression was observed in the mesenchymal clusters, whereasWnt10b expression was observed in both mesenchymal and epithelial clusters (Fig. S6). However, because only a small fraction of cells expressed Wnt10b, the principal cellular source of WNT10B protein remains unclear. Therefore, defining the precise spatial patterns of Wnt10b and Fgf2 in regular regeneration will be an important goal for future work.

Data availability:I assume that the RNA-sequencing data will be deposited at a public repository.

RNA-seq FASTQ files have been deposited in the DNA Data Bank of Japan (DDBJ; https://www.ddbj.nig.ac.jp/) under BioProject accession PRJDB38065. We have added a Data availability section to the revised manuscript.

References

Castilla-Ibeas, A., Zdral, S., Oberg, K. C., & Ros, M. A. (2024). The limb dorsoventral axis: Lmx1b’s role in development, pathology, evolution, and regeneration. Developmental Dynamics, 253(9), 798–814. https://doi.org/10.1002/dvdy.695

Johnson, G. L., Glasser, M. B., Charles, J. F., Duryea, J., & Lehoczky, J. A. (2022). En1 and Lmx1b do not recapitulate embryonic dorsal-ventral limb patterning functions during mouse digit tip regeneration. Cell Reports, 41(8), 111701. https://doi.org/10.1016/j.celrep.2022.111701

Stocum, D. (2017). Mechanisms of urodele limb regeneration. Regeneration, 4. https://doi.org/10.1002/reg2.92

Tank, P. W., & Holder, N. (1978). The effect of healing time on the proximodistal organization of double-half forelimb regenerates in the axolotl, Ambystoma mexicanum. Developmental Biology, 66(1), 72–85. https://doi.org/10.1016/0012-1606(78)90274-9